# Efficient Vocabulary-Free Fine-Grained Visual Recognition in the Age of Multimodal LLMs

**Hari Chandana Kuchibhotla**  *ai20resch11006@iith.ac.in*
*Indian Institute of Technology Hyderabad, India*

**Sai Srinivas Kancheti**  *cs21resch01004@iith.ac.in*
*Indian Institute of Technology Hyderabad, India*

**Abbavaram Gowtham Reddy**  *gowtham.abbavaram@cispa.de*
*CISPA Helmholtz Center for Information Security, Saarbrücken, Germany*

**Vineeth N Balasubramanian**  *vineeth.nb@microsoft.com & vineeth.nb@cse.iith.ac.in*
*Microsoft Research India & Indian Institute of Technology Hyderabad, India*

**Reviewed on OpenReview:** *https://openreview.net/forum?id=FvA0UMw9X2*

## Abstract

Fine-grained Visual Recognition (FGVR) involves distinguishing between visually similar categories, which is inherently challenging due to subtle inter-class differences and the need for large, expert-annotated datasets. In domains like medical imaging, such curated datasets are unavailable due to issues like privacy concerns and high annotation costs. In such scenarios lacking labeled data, an FGVR model cannot rely on a predefined set of training labels, and hence has an unconstrained output space for predictions. We refer to this task as Vocabulary-Free FGVR (VF-FGVR), where a model must predict labels from an unconstrained output space without prior label information. While recent Multimodal Large Language Models (MLLMs) show potential for VF-FGVR, querying these models for each test input is impractical because of high costs and prohibitive inference times. To address these limitations, we introduce **Nea**rest-Neighbor Label **R**efinement (NeaR), a novel approach that fine-tunes a downstream CLIP model using labels generated by an MLLM. Our approach constructs a weakly supervised dataset from a small, unlabeled training set, leveraging MLLMs for label generation. NeaR is designed to handle the noise, stochasticity, and open-endedness inherent in labels generated by MLLMs, and establishes a new benchmark for efficient VF-FGVR.

## 1 Introduction

Fine-Grained Visual Recognition (FGVR) is a task in computer vision that focuses on distinguishing between highly similar categories within a broader class (Wei et al., 2021). This task has gained increased importance in recent years given the success of foundation models on coarse-grained classification tasks. Traditional image classification aims to differentiate between dogs, cats, and birds, while FGVR aims to distinguish between different subcategories, such as *Tennessee Warbler, Yellow-rumped Warbler* and *Orange-crowned Warbler* among birds as an example. Fine-grained understanding is essential for a wide range of applications, including biodiversity studies (Liu et al., 2024b; Yang et al., 2020), medical diagnosis (Ridzuan et al., 2022; Sohoni et al., 2020), manufacturing (Yang et al., 2022), fashion retail (Cheng et al., 2021), and agriculture (Yang et al., 2020).

FGVR poses significant challenges due to the subtle differences between categories and the need for large annotated datasets. Typically, fine-grained classification datasets are annotated by domain experts who

meticulously examine each image and assign a corresponding label. However, when such domain experts are unavailable, not only is labeled data unavailable, but the underlying fine-grained categories of interest may also be unknown. We refer to this task as Vocabulary-Free FGVR (VF-FGVR), where the vocabulary of fine-grained labels is not provided. In this context, pre-trained Multimodal Large Language Models (MLLMs), which are trained on vast corpora of image-text data and possess extensive world knowledge, provide a contemporary solution (Reid et al., 2024). MLLMs excel at zero-shot multimodal tasks such as Visual Question Answering (VQA). By recasting FGVR as a VQA problem, where we ask the question 'What is the best fine-grained class label for this image?', MLLMs can predict fine-grained labels in a zero-shot manner without prior training on a specific dataset. The ability of MLLMs to operate in a VF setting presents a promising approach for domains where curated, labeled datasets are scarce or unavailable.

However, querying such large models for each test input is computationally expensive and time-consuming, making their usage impractical at scale. As shown in Table 1, performing inference with GPT-4o (Achiam et al., 2023) on $32,203$ test images from benchmark FGVR datasets takes $\approx 17.5$ hours of querying and incurs a cost of $\approx$ USD \$100.

*Hence, especially considering the need for sustainable AI systems, there is an imminent need for an efficient VF-FGVR system that performs on par with MLLMs on the fine-grained understanding task, while being efficient in terms of time and computational resources required.* We focus on this problem in this work. To this end, we only consider access to a small unlabeled set of training images belonging to the classes of interest, with no information regarding the individual class names or even the total number of classes. Such unlabeled

| Method | Training Time | Inference Time | Cost (\$) | cACC |
|---|---|---|---|---|
| FineR (Liu et al., 2024a) | 10 $h$ | 1.12 $h$ | 0 | 57.0 |
| [†] GPT-4o | - | 17.5 $h$ | $\sim 100$ | 59.2 |
| ZS-CLIP + GPT-4o | - | 0.03 h | 1 | 54.6 |
| NeaR + GPT-4o (Ours) | 1.57 $h$ | 0.03 h | 1 | **67.6** |
| [‡] LLaMA | - | 9.12 $h$ | 0 | 48.4 |
| ZS-CLIP + LLaMA | - | 0.03 $h$ | 0 | 60.5 |
| NeaR + LLaMA (Ours) | 1.57 $h$ | 0.03 $h$ | 0 | **65.0** |

Table 1: Performance and cost metrics for different methods on benchmark FGVR datasets (computed over $32,503$ images from the test sets). † = proprietary models, ‡ = open-source models (both used only for inference). Our method NeaR achieves a clustering accuracy (cACC described in § 4) that exceeds even direct MLLM queries, at a fraction of cost and time taken.

datasets are relatively easy to obtain for many domains – for instance, a collection of unlabeled photos of exotic birds. We empirically show in § 4 that having about 3 images per class is sufficient to learn an FGVR system that can perform inference for any number of test samples in the considered experiments. To solve this VF-FGVR task, we propose to label this training set by querying an MLLM for each image. Such a noisily labeled training set can be used in different ways – a simple strategy would be to utilize this MLLM supervision to build a zero-shot classifier over the set of generated labels using a pre-trained CLIP (Radford et al., 2021) model. Another way would be to naively fine-tune the CLIP model using the generated labels. In both these approaches, test images can now be classified by the CLIP model without the need for expensive forward passes through an MLLM. Although such simple methods are efficient in compute and time taken, they fall short on performance as shown in § 4. This is because MLLM outputs are inherently noisy and open-ended, so the generated labels do not necessarily provide strong supervision.

To address these limitations, we propose **Nea**rest-Neighbor Label **R**efinement (**NeaR**), a method designed to learn using the noisy labels generated by an MLLM. Our approach first constructs a candidate label set for each image using the generated labels of other similar images. In line with prior work on learning with noisy labels (LNL) (Li et al., 2020), we partition the dataset into clean and noisy samples. We then design a label refinement scheme for both partitions that can effectively combine information from the constructed candidate set and the generated label. Finally, to address the open-ended nature of MLLM outputs, we incorporate a label filtering mechanism to truncate the label space. Our method NeaR thus enables us to handle the inherently noisy and open-ended labels generated by MLLMs, allowing us to effectively fine-tune a downstream CLIP model. As shown in Table 1, for GPT-4o, our approach can achieve performance exceeding that of direct inference while incurring only $1/100^{th}$ of the total inference cost, and requiring a negligible fraction of inference time.

Our key contributions can be summarized as: **(i)** Differing from existing works, we study how contemporary models can be used to build a cost-efficient vocabulary-free fine-grained visual recognition system, **(ii)** We propose a pipeline that can handle noisy and open-ended labels generated by an MLLM. Our proposed

method NeaR leverages similarity information to construct a candidate label set for each image which is used to mitigate the impact of label noise. We also design a label filtering mechanism to improve classification performance. **(iii)** We perform a comprehensive set of experiments showing that NeaR outperforms existing works and the MLLM-based baselines we introduce for VF-FGVR (detailed in the baselines paragraph of § 4), achieving this in a cost-efficient way.

## 2  Related Work

**Fine-Grained Visual Recognition.** FGVR (Wah et al., 2011; Maji et al., 2013) aims to identify sub categories of an object, such as various bird species, aircraft type etc. FGVR has been extensively studied in prior work (Wei et al., 2021). A key limitation of these methods is their reliance on annotated datasets, which are often unavailable in many important domains like e-commerce and medical data. With advancements in Vision-Language Models and MLLMs, the burden of dataset annotation can be alleviated, reducing the need for extensive human effort.

**Foundation Models for VF-FGVR.** Recent advancements in MLLMs have led to models demonstrating strong zero-shot performance across a wide range of multimodal tasks (Li et al., 2023; Achiam et al., 2023; Reid et al., 2024; Touvron et al., 2023; Liu et al., 2023). These MLLMs can be applied to VF-FGVR by framing the task as a VQA problem. MLLMs are broadly categorized into two types: (1) Proprietary models, such as GPT-4o (Achiam et al., 2023) and GeminiPro (Reid et al., 2024), and (2) Open-source models, including BLIP-v2 (Li et al., 2023), LLaVA-1.5-7B (Liu et al., 2023), LLaMA-3.2-11B (Touvron et al., 2023) and Qwen2-7B (Wang et al., 2024). Recent works (He et al., 2025; Zhang et al., 2024) examine the zero-shot fine-grained performance of such MLLMs in the closed-world setting and show that finetuning can help. Such approaches are not applicable in the VF-FGVR setup where labeled data as well as the label space, is unavailable. As shown in Table 1, for both types of MLLMs, performing inference for every test point remains computationally expensive and time-consuming. To address this, recent works have developed more efficient solutions for VF-FGVR. For instance, (Liu et al., 2024a;c; Conti et al., 2023) propose pipelines that use cascades of MLLMs. FineR (Liu et al., 2024a) presents a pipeline combining VQA systems, Large Language Models (LLMs), and a downstream CLIP model, leveraging unsupervised data to build a multimodal classifier for inference. RAR (Liu et al., 2024c) uses a multimodal retriever with external memory, retrieving and ranking top-k samples using an LLM. RAC (Long et al., 2022) uses retrieval to refine features via a learned transformer using a labeled memory bank, operating under full supervision. In contrast with RAC, NeaR is lightweight, backbone-agnostic, and avoids complex memory-based architectures. Furthermore both RAR and RAC operate under full supervision, making them unsuitable for the VF-FGVR task. CaSED (Conti et al., 2023) approaches VF-FGVR by accessing an external database to retrieve relevant text for a given image. Nevertheless, these methods are often complex and do not fully exploit the advancements in MLLMs, resulting in suboptimal performance.

**Prompt Tuning.** Prompt-tuning methods add a small number of learnable tokens to the input while keeping the pretrained parameters unchanged. The tokens are fine-tuned to enhance the performance of large pre-trained models on specific tasks. Context Optimization (CoOp) (Zhou et al., 2021) was the first to introduce text-based prompt tuning, replacing manually designed prompts like *"a photo of a"* with adaptive soft prompts. We study the impact of our method under other prompt-tuning methods such as VPT (Jia et al., 2022) and IVLP (Rasheed et al., 2022) in Appendix§ A8.5.

**Learning with Noisy Labels.** (Arpit et al., 2017) demonstrated the memorization effect of deep networks, showing that models tend to learn clean patterns before fitting noisy labels. To mitigate this, (Han et al., 2018; Chen et al., 2019) introduce iterative learning methods to filter out noisy samples during training. (Arazo et al., 2019) proposed a mixture model-based approach to partition datasets into clean and noisy subsets, leading to more reliable training. Building on these insights, DivideMix (Li et al., 2020), a state-of-the-art LNL method, combines semi-supervised learning with data partitioning to achieve superior performance on noisy datasets. JoAPR (Guo & Gu, 2024) is a contemporary approach to fine-tune CLIP on noisy few-shot data. We compare against JoAPR in Tab 4.

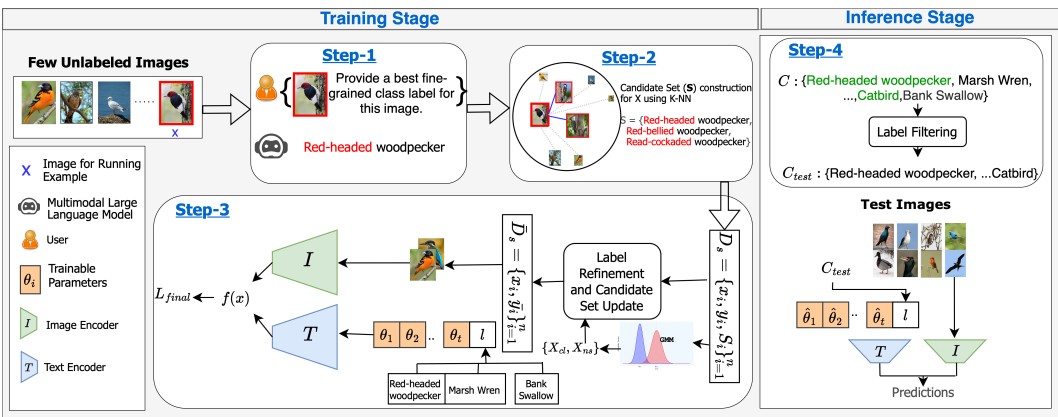

Figure 1: Overview of our proposed method, **NeaR**, for Vocabulary-Free Fine-Grained Visual Recognition (VF-FGVR). In the Training Stage, we start with a few unlabeled images. Step-1: An MLLM generates a best-estimate fine-grained label (e.g., "Red-headed woodpecker") for each image. Step-2: A candidate label set is constructed using K-Nearest Neighbors, capturing related fine-grained classes. Step-3: The model is fine-tuned using a CLIP-based architecture. A GMM is applied to the loss to partition the data into clean and noisy samples. Based on this split, a label refinement mechanism is used to further update and refine the labels. The final loss, $L_{final}$, is then computed, and the model parameters are updated accordingly. In the Inference Stage, we apply label filtering to limit the label space (Step-4). Our approach handles the noise and open-ended nature of MLLM-generated labels, significantly reducing inference time and cost while maintaining performance.

## 3 Methodology

As shown in Table 1, although MLLMs are capable of performing VF-FGVR, labeling every test image is expensive and time consuming which limits their practical application. To address these limitations, we propose **Nea**rest-Neighbor Label **R**efinement (**NeaR**), a method designed to leverage MLLMs efficiently for VF-FGVR. Our approach begins by constructing a candidate label set for each unlabeled training image as described in § 3.2. Next we partition the data into 'clean' and 'noisy' samples using a Gaussian Mixture Model (GMM) and the small-loss rule (Arpit et al., 2017). In § 3.3, we show how to refine labels of noisy samples by incorporating information from the candidate sets. The final loss function is a simple cross-entropy loss, where the refined labels serve as the targets. As outlined in § 3.1, using MLLM outputs directly can result in an excessively large label space, which can hinder performance. To address this, NeaR incorporates a label filtering mechanism to truncate the label space, which boosts classification performance while improving efficiency. Once the CLIP model is fine-tuned, it can classify test images without requiring additional expensive forward passes through an MLLM, significantly improving inference efficiency. Our methodology NeaR is able to exceed the performance of direct MLLM-based classification at just a fraction of the cost and compute. An overview of our methodology is presented in Figure 1, and the pseudocode is detailed in Algorithm§ 1 in the appendix. We begin by discussing the necessary preliminaries.

### 3.1 Preliminaries

**Problem Formalization.** We consider a setting where only a small, unlabeled training set of $n$ images $X = \{x_i\}_{i=1}^n$, $x_i \in \mathcal{X}$ is available. We assume that this training set has at least $m$-shot samples for each class of the unknown ground-truth class name set $\mathcal{G}$. We also study a more realistic scenario where there is class imbalance in § 4.2 and observe that the performance of NeaR does not degrade. Note that no further information about $\mathcal{G}$ is known, including its cardinality. For each image $x_i$, we obtain a class name $l_i = L(x_i, p)$ from an MLLM $L$, where $p$ is a simple text prompt 'Provide a best fine-grained class name for this image.' that guides the MLLM to generate a fine-grained class name for the image. The generated dataset $\mathcal{D} = \{(x_i, l_i)\}_{i=1}^n$ consists of $n$ image-label pairs, and the output space of class names is

denoted by $\mathcal{C} = \bigcup_{i=1}^{n} l_i$. W.l.o.g. we assume $\mathcal{C}$ is lexicographically ordered and we denote the $k = |\mathcal{C}|$ labels by $\mathcal{C} = \{c_1, c_2, \ldots, c_k\}$. Let $y_i \in \{0,1\}^k$, $i \in [n]$, denote the one-hot encoding of the text label $l_i$ for image $x_i$ i.e $y_i^j = 1$ if $l_i = c_j$ and 0 otherwise. Due to the inherent noise and stochastic nature of MLLM generated labels, it is common to have $|\mathcal{C}| > |\mathcal{G}|$. Furthermore, the number of generated labels increases with the size of the training set and can become prohibitively large, hampering training and reducing efficiency of the downstream CLIP classifier.

**CLIP Classifier:** CLIP (Radford et al., 2021) consists of an image encoder $\mathcal{I}$ and a text encoder $\mathcal{T}$ trained contrastively on image-text pairs. For VF-FGVR, we first query an MLLM to build a few-shot dataset $\mathcal{D}$ with label space $\mathcal{C}$. CLIP classifies image $x$ as $\hat{l}(x) = \arg\max_{l \in \mathcal{C}} cos(\mathcal{I}(x), \mathcal{T}(l))$. We call this baseline ZS-CLIP. Note that ZS-CLIP does not leverage the paired supervision in $\mathcal{D}$ and thus serves as a simple baseline. CLIP can benefit from fine-tuning on a small labeled datast. CoOp (Zhou et al., 2021) is a prompt-tuning approach that adds a small number of learnable tokens $\theta$ to the class name. We denote CLIP's class predictions by $f_\theta(x) \in \Delta^k$, where $\Delta^k$ denotes the $k-1$ simplex. The prompts are trained on the dataset $\mathcal{D}$ by minimizing cross-entropy loss $L_{CE}(\theta) = \frac{-1}{n} \sum_{i=1}^{n} \sum_{j=1}^{k} y_i^j log(f_\theta^j(x_i))$. Naive finetuning using CoOp on a dataset generated by an MLLM can be susceptible to label noise. Our NeaR method mitigates this issue by refining labels through nearest-neighbor information. The following sections detail how NeaR constructs candidate sets, learns with these sets, and filters labels for effective downstream performance.

## 3.2 NeaR: Candidate Set Construction

The label generated by an MLLM in response to a prompt may vary significantly from the ground truth class label for each image. We propose to leverage local geometry to mitigate the noise in generated labels. More formally, we make the *manifold assumption*, which suggests that similar images should share similar or identical class labels (Iscen et al., 2022; Li et al., 2022). This is particularly useful when the label $l_i$ assigned to image $x_i$ by the MLLM is incorrect, which we refer to as a noisy label. By constructing a candidate label set, we increase the likelihood of including the true label or a semantically closer alternative in the candidate set rather than relying solely on the potentially incorrect label provided by the MLLM.

Table 2 shows that the semantic similarity between the best label in the candidate set and the ground truth is higher than that of the noisy single label for the training images, supporting our hypothesis. We use CLIP's pretrained image encoder $\mathcal{I}$ to extract image features of the entire training set $X$. For each image $x_i$, we select the top-$\kappa$ most similar images (including $x_i$ itself) and gather their corresponding labels to form the candidate set $S_i = (l_i, l_1, \ldots, l_{\kappa-1})$. In this work, we choose $\kappa = 3$. The resulting dataset is reconstructed as $\mathcal{D}_s = \{(x_i, l_i, S_i)\}_{i=1}^{n}$, incorporating the candidate sets rather than single labels alone. An alternative way of noise mitigation is to have the MLLM directly generate a candidate set of 'top-$\kappa$'

| Method | Bird-200 | Car-196 | Dog-120 | Flower-102 | Pet-37 |
|---|---|---|---|---|---|
| MLLM Labels | 69.9 | 78.6 | 70.9 | 57.9 | 84.8 |
| Random CS | 72.8 | 78.9 | 73.6 | 61.6 | 85.9 |
| K-NN CS (ours) | **78.0** | **81.3** | **75.5** | **65.8** | **87.4** |

Table 2: The table compares the quality of labels generated by the MLLM (LLaMA), a Random candidate set (CS) where we pick 2 other labels (for $\kappa = 3$) at random from the set of all MLLM generated labels, and K-NN CS using sACC. While Random CS modestly increases the likelihood of including the true label compared to using MLLM labels directly, K-NN CS significantly outperforms Random CS, generating a superior candidate set and validating our hypothesis.

labels for each image, instead of just a single label. However this approach does not make use of similarity information between images, as each candidate set is now generated independently, leading to an excessively large label space. We empirically demonstrate the effectiveness of our nearest-neighbor based candidate set generation over other alternatives in Appendix § A8.4.

## 3.3 NeaR: Learning With a Candidate Set

We treat the candidate sets as a source of supplementary similarity information to be used in conjunction with the label. As explained below, for a *noisy* image, where the initial label $l_i$ is incorrect, we propose

relying on the candidate set $S_i$ to mitigate the impact of noise. Conversely, for a *clean* image, we can trust and utilize its generated label $l_i$. **Detecting Noisy Samples.** It has been demonstrated in (Arpit et al., 2017) that models tend to learn clean samples before noisy ones, resulting in lower loss values for clean samples. Following DivideMix (Li et al., 2020), for every training epoch, we fit a two-component Gaussian Mixture Model (GMM) over the cross entropy loss values of all training samples $\{L(x_i, l_i)\}_{i=1}^n$, where $L(x_i, l_i) = -\sum_{j=1}^k y_i^j log(f_\theta^j(x_i))$, $y_i$ is the one-hot encoding of $l_i$. The component with the smaller mean value models the clean samples, while the other component models the noisy ones. The posterior probability $w_i = \mathbb{P}_{GMM}(clean|x_i)$ computed from the fitted GMM is used to model the likelihood that a sample $x_i$ is clean. This GMM is refitted for every training epoch, enabling dynamic estimation of label noise over time. We now partition the training data into clean $X_{cl} = \{x_i \in X \mid w_i \geq \tau\}$ and noisy $X_{ns} = X \setminus X_{cl}$ sets based on clean probability threshold $\tau$. We use the average clean posterior as an adaptive threshold for every epoch, i.e $\tau = \frac{1}{n} \sum_{i=1}^n w_i$. The effect of different thresholding strategies is presented in Appendix § A8.6.

**Warm-up.** Warm-up strategies are commonly used to speedup convergence and stabilize training. As demonstrated in (Li et al., 2020), an initial warm-up phase allows the model to learn the clean samples better, resulting in better separation between the losses of clean and noisy samples. During warm-up, we train prompts for a few epochs (10 in our experiments) by minimizing the cross entropy loss over the generated labels $l_i$ with one-hot representation $y_i$:

$$L_{warmup}(\theta) = \frac{-1}{n} \sum_{i=1}^n \sum_{j=1}^k y_i^j \cdot log(f_\theta^j(x_i))$$

This warm-up step lays the groundwork for effective training by allowing the model to initially focus on images labeled correctly by the MLLM.

**Candidate Set Guided Label Refinement.** Following the initial warm-up phase, we make a forward pass over the entire training set at each training epoch to fit a GMM and partition data into clean and noisy samples $X_{cl}$ & $X_{ns}$ as described earlier. We model the confidence of the candidate set $S_i$ by a vector $q_i \in \mathbb{R}^k$ for each image $i \in [n]$, initialized as $q_i^j = \frac{1}{|S_i|}$ if $c_j \in S_i$ and 0 otherwise. This initialization reflects uniform confidence over classes belonging to the candidate set, and zero for non-members. A candidate set is derived from neighboring images and provides a broader view of possibly correct labels. Our approach constructs refined labels to effectively leverage this additional information. We propose to construct refined labels for clean and noisy images differently. For an image $x_i$ with one-hot label $y_i$ and candidate set confidence $q_i$, we construct a refined label $\bar{y}_i$ by performing label-mixup using model predictions $f_\theta$ as:

$$\bar{y}_i = \begin{cases} \text{shrp}\big(w_i \cdot y_i + (1 - w_i) \cdot f_\theta(x_i), T\big), & \text{if } x_i \in X_{cl} \\ \text{rsc}\big(\text{shrp}(w_i \cdot q_i + (1 - w_i) \cdot f_\theta(x_i), T), q_i\big), & \text{o/w} \end{cases}$$

where $w_i$ is the GMM clean posterior probability, and $f_\theta(x_i)$ denotes the CLIP model class probabilities with learnable prompts $\theta$. The label-mixup is conditioned on clean posterior probability $w_i$, enabling the model to rely on the pseudo-label when $w_i$ is small. The sharpen function $shrp(y, T)^i = (y^i)^{\frac{1}{T}} / \sum_{j=1}^k (y^j)^{\frac{1}{T}}$, as defined in (Berthelot et al., 2019), adjusts a probability distribution $y$ to be more confident using a temperature $T$. The rescale function is $rsc(y, q)^i = (y \odot q)^i / \sum_{j=1}^k (y \odot q)^j$, where $\odot$ represents the hadamard product, rescales a probability $y$ with the current confidence estimates of the candidate set $q$. This ensures that the refined label has non-zero probabilities only for the candidate labels. For both clean & noisy images, we update candidate set confidence to be used in the next epoch as $q_i = \text{rsc}(f_\theta(x_i), 1[q_i])$ where $1[q_i]$ is 1 at non-zero indices. Prompts are learned by minimizing the cross-entropy loss between the refined labels $\bar{y}$ and CLIP model output.

$$L_{final}(\theta) = \frac{-1}{n} \sum_{i=1}^n \sum_{j=1}^k \bar{y}_i^j \cdot log(f_\theta^j(x_i))$$

| Method | Bird-200 | | Car-196 | | Dog-120 | | Flower-102 | | Pet-37 | | Average | |
|---|---|---|---|---|---|---|---|---|---|---|---|---|
| | cACC | sACC | cACC | sACC | cACC | sACC | cACC | sACC | cACC | sACC | cACC | sACC |
| CoOp-GT (Upper Bound) | 58.1 | 81.1 | 65.1 | 66.6 | 64.0 | 79.9 | 74.6 | 82.1 | 88.2 | 92.5 | 70.0 | 80.4 |
| CaSED | 25.6 | 50.1 | 26.9 | 41.4 | 38.0 | 55.9 | 67.2 | 52.3 | 60.9 | 63.6 | 43.7 | 52.6 |
| FineR | 51.1 | 69.5 | 49.2 | 63.5 | 48.1 | 64.9 | 63.8 | 51.3 | 72.9 | 72.4 | 57.0 | 64.3 |
| RAR | 51.6 | 69.5 | 53.2 | 63.6 | 50.0 | 65.2 | 63.7 | 53.2 | 74.1 | 74.8 | 58.5 | 65.3 |
| [†]GPT-4o | 68.8 | 85.2 | 37.4 | 61.5 | 71.1 | 80.4 | 50.5 | 51.6 | 68.2 | 83.5 | 59.2 | 72.4 |
| ZS-CLIP-GPT4o | 48.8 | 72.5 | 42.9 | 59.5 | 43.8 | 69.1 | 18.2 | 53.0 | 68.2 | 78.7 | 54.6 | 66.6 |
| CoOp-GPT-4o | 54.4 | 75.4 | 51.9 | 59.8 | 60.4 | 72.9 | 70.4 | 51.7 | 83.5 | 86.3 | 64.1 | **69.2** |
| NeaR-GPT4o | 55.8 | 75.6 | 57.0 | 60.0 | 61.6 | 74.4 | 80.6 | 52.1 | 82.9 | 84.0 | **67.6(+3.5%)** | **69.2** |
| [†]Gemini Pro | 66.1 | 82.7 | 35.4 | 62.8 | 65.8 | 81.2 | 45.3 | 54.3 | 71.3 | 85.7 | 56.8 | 73.3 |
| ZS-CLIP-GeminiPro | 51.7 | 74.6 | 41.6 | 61.7 | 58.9 | 72.6 | 57.7 | 49.1 | 71.7 | 78.6 | 56.3 | 67.3 |
| CoOp-GeminiPro | 55.2 | 75.9 | 50.2 | 61.5 | 62.7 | 73.8 | 68.3 | 51.2 | 81.6 | 83.8 | 63.6 | 69.2 |
| NeaR-GeminiPro | 55.9 | 76.0 | 54.9 | 61.1 | 64.7 | 75.4 | 77.9 | 53.2 | 79.4 | 80.8 | **66.6(+3%)** | **69.3(+0.1%)** |
| [‡]Qwen2-VL-7B-Instruct | 53.0 | 75.3 | 45.6 | 63.7 | 69.7 | 78.8 | 84.8 | 72.7 | 77.7 | 85.1 | 66.2 | 75.1 |
| ZS-CLIP-Qwen2 | 41.0 | 66.0 | 50.8 | 60.8 | 59.3 | 70.5 | 66.7 | 55.2 | 72.4 | 77.2 | 58.0 | 65.9 |
| CoOp-Qwen2 | 51.0 | 72.1 | 52.1 | 61.9 | 62.5 | 73.4 | 77.0 | 65.0 | 83.4 | 87.5 | 65.2 | **72.0** |
| NeaR-Qwen2 | 48.9 | 72.0 | 55.6 | 63.2 | 62.0 | 73.3 | 81.4 | 68.0 | 84.6 | 86.8 | **66.5(+1.3%)** | 71.7(-0.3%) |
| [‡]LLaMA-3.2-11B | 41.4 | 70.6 | 14.4 | 61.6 | 55.0 | 71.8 | 66.0 | 63.6 | 65.1 | 82.0 | 48.4 | 69.9 |
| ZS-CLIP-LLaMA | 48.7 | 66.3 | 45.8 | 60.6 | 57.4 | 65.9 | 74.8 | 58.4 | 76.0 | 78.4 | 60.5 | 65.9 |
| CoOp-LLaMA | 49.2 | 68.7 | 45.5 | 60.7 | 58.4 | 68.4 | 75.9 | 59.8 | 74.4 | 79.2 | 60.7 | 67.4 |
| NeaR-LLaMA | 51.0 | 70.2 | 52.6 | 60.9 | 59.2 | 70.2 | 78.6 | 61.7 | 83.5 | 86.2 | **65.0(+4.3%)** | **69.8(+2.4%)** |

Table 3: ZS-Zero Shot, [†] proprietary models used for inference, [‡] open-source models used for inference. Our results shown here are for $\kappa = 3$ and $m = 3$. The first row is CoOp-GT, where we finetune a CLIP model using ground-truth labels , serving as an upper bound. The second partition consists of contemporary VF baselines of which FineR (Liu et al., 2024a) is best performing. We outperform FineR by a large margin, even when using weaker open-source MLLMs. The next four partitions are for labels generated by various MLLMs. We compare NeaR against CoOp within each partition, and highlight best numbers in **bold**. Our method NeaR outperforms all contemporary baselines, as well as ZS-CLIP and CoOp baselines for a variety of MLLMs.

**Connection to PRODEN.** Our loss is similar in spirit to losses designed for Partial Label Learning (PLL), such as PRODEN (Feng et al., 2020), which allow learning when only candidate labels are present. However unlike the PLL setting, our candidate sets are constructed for every image using noisy MLLM outputs, and may not contain the true label. Furthermore, our method uniquely benefits from access to an initial 'best-estimate' label $l_i$ generated by the MLLM, which is not exploited by traditional PLL algorithms. This best estimate label allows us to differentiate clean samples and helps training convergence by transferring knowledge from clean to noisy samples through iterative updates of $q$.

### 3.4 NeaR: Label Filtering

Although we train on the entire label set $\mathcal{C}$, we observe that many labels are noisy and can be removed from the inference time label space. Let $F_{clip} = \{c_i \mid \exists x \in X \text{ s.t } i = \arg\max_{j \in [k]} f_\theta^j(x)\}$ be a filtered set of labels which are predicted by CLIP on the training set. Let $F_{cand} = \{c_i \mid \text{ s.t } i = \arg\max_{j \in [k]} q_i^j\}$ be another filtered set of labels which are predicted using just the candidate sets. We propose to keep only those labels which belong to both sets. The evaluation time label space is $\mathcal{C}_{test} = F_{clip} \cap F_{cand}$ and the inference time prediction of an image $x$ is $\hat{l}(x) = \arg\max_{l \in \mathcal{C}_{test}} sim(\mathcal{I}(x), \mathcal{T}_{\hat{\theta}}(l))$, where $\hat{\theta}$ are the learned prompts. Label filtering is effective as shown in Table A14.

## 4 Experiments and Results

In this section, we comprehensively evaluate the classification performance of NeaR for the VF-FGVR task. We begin by describing the datasets, metrics and benchmark methods we compare against.

**Datasets:** We perform experiments on five benchmark fine-grained datasets: CaltechUCSD Bird-200 (Wah et al., 2011), Stanford Car-196 (Khosla et al., 2011), Stanford Dog-120 (Krause et al., 2013), Flower-102 (Nilsback & Zisserman, 2008), Oxford-IIIT Pet-37 (Parkhi et al., 2012). Following (Liu et al., 2024a), for each dataset, NeaR and other baselines only have access to $m$ unlabeled training images per class. Unless specified otherwise, we assume $m = 3$. Results for $1 \leq m \leq 10$ are shown in Figure 2.

**Baselines:** We compare our method NeaR against four different classes of baseline methods. **(i)** Direct Inference on MLLMs. For every test image, we directly query an MLLM for a fine-grained label using a text prompt such as 'What is the best fine-grained class name for this image?'. We evaluate two proprietary MLLMs – GPT-4o (Achiam et al., 2023) and GeminiPro (Reid et al., 2024) and two strong open-source MLLMs, LLaMA-3.2-11B-Vision-Instruct (Touvron et al., 2023) and Qwen2-VL-7B-Instruct (Wang et al., 2024). In the Appendix § A8, we show results on two other weaker open-source MLLMs, BLIP-2 (Li et al., 2023) and LLaVA-1.5 (Liu et al., 2023). **(ii)** Contemporary VF Baselines. We consider three contemporary baselines which do not require expert annotations but use foundational models to perform VF-FGVR – CaSED (Conti et al., 2023), FineR (Liu et al., 2024a) and RAR (Liu et al., 2024c). **(iii)** ZS-CLIP with MLLM label space. As described in § 3.1, we can perform zero-shot classification using pre-trained CLIP over the label space generated by querying various MLLMs on training images. We consider four variants of ZS-CLIP – ZS-CLIP-GPT4o, ZS-CLIP-GeminiPro, ZS-CLIP-LLaMA, and ZS-CLIP-Qwen2. **(iv)** Prompt Tuning Baselines. Following CoOp as described in § 3.1, we directly perform prompt-tuning using the labels generated by an MLLM. We consider four variants – CoOp-GPT4o, CoOp-GeminiPro, CoOp-LLaMA, and CoOp-Qwen2.

**Evaluation Metrics:** In the VF-FGVR setting, NeaR as well as all other baselines operate in an unconstrained label space, making accuracy an invalid metric since the predicted labels may never exactly match the ground-truth labels. Following FineR (Liu et al., 2024a) and CaSED (Conti et al., 2023), we evaluate performance using two complementary metrics: *Clustering Accuracy* (cACC) and *Semantic Accuracy* (sACC). cACC measures the ability of the model to group similar images together. For $M$ test images with ground-truth labels $y^{\star}$ and predicted labels $\hat{y}$, cACC is computed as $\max\limits_{p \in \mathcal{P}(\hat{\mathcal{Y}})} \frac{1}{M} \sum\limits_{i=1}^{M} \mathbb{1}(y_i^{\star} = p(\hat{y}_i))$, where $\mathcal{P}(\hat{\mathcal{Y}})$ is the set of all permutations of the generated labels. Since cACC disregards the actual label name, it does not measure if the predictions are semantically correct. Despite this limitation, cACC is a strong evaluation metric and is widely used in areas such as GCD (Vaze et al., 2022), where the goal is to assess consistency of predictions rather than exact label semantics. Semantic closeness is captured by sACC, which measures the cosine similarity between Sentence-BERT (Reimers & Gurevych, 2019) embeddings of the predicted and ground-truth labels. As observed in (Liu et al., 2024a), sACC is a more forgiving metric than cACC, because embedding based similarity methods can capture general semantics even for completely distinct labels. We hence consider cACC as representative of the model's performance, with sACC acting as a sanity check to ensure that the predicted labels remain meaningful.

**Implementation Details:** We use CLIP ViT-B/16 (Radford et al., 2021) as the VLM, whose image encoder we also use to find the $\kappa$-nearest neighbors, with $\kappa = 3$ by default. We strictly follow FineR(Liu et al., 2024a) for the default number of shots i.e $m = 3$ and for the few-shot training splits. For both the CoOp baseline and our method, we introduce 16 trainable context vectors. The same set of prompts are optimized during the warmup stage, and for the subsequent training stage. We use SGD as the optimizer and train for 50 epochs, with 10 warmup epochs. We use a temperature of 2 in the sharpening function. Our batch size is 32. We use the SGD optimizer with a learning rate of 0.002, and use both constant learning rate scheduler and cosine annealing scheduler sequentially. The training hyperparameters are the same for CoOp and NeaR. We sample an equal batch of clean and noisy samples during every epoch. We run all our experiments on a single Nvidia Tesla V100-32GB GPU with an Nvidia driver version of 525.85.12. We use PyTorch 2.4.0 and CUDA 12.0. We utilize the publicly available meta-llama/Llama-3.2-11B-Vision-Instruct model and Qwen/Qwen2-VL-2B-Instruct model from HuggingFace. We observe that instruction tuned MLLMs generate better labels compared to base models. We perform inference using the HuggingFace `transformers`

library (Wolf et al., 2019). Unless otherwise specifically stated, we use LLaMA-3.2 as our default MLLM. Our code is available at **https:/github.com/NeaR**.

## 4.1 Main Results

In this section we compare NeaR against baselines on five fine-grained datasets. In addition to the considered baselines, we benchmark against JoAPR (Guo & Gu, 2024), a state-of-the-art noisy label learning method designed for CLIP, and against PRODEN (Feng et al., 2020), a widely used partial label learning algorithm.

**Benchmarking NeaR Against Baseline Methods:** We evaluate NeaR against the four categories of baselines introduced in § 4 – Direct MLLM inference, contemporary VF methods, zero-shot CLIP, and prompt-tuned CLIP. The results are shown in Table 3, with all numbers reported for 3-shot training images. The first partition of the table, CoOp-GT, is the performance of fine-tuned CLIP model when provided with the ground-truth label space, serving as an upper bound. The next partition consists of contemporary methods that can perform VF-FGVR. Out of these, FineR (Liu et al., 2024a) is conceptually closest to ours as it uses a combination of an LLM and a VQA system to construct a training-free CLIP based classifier. We outperform FineR on all datasets by a margin of at least +**8**% in average cACC, even when using labels from open-source MLLMs.

Moreover, as shown in Table 1, NeaR is significantly more efficient in terms of computation time. The next four partitions in Table 3 report results using labels generated by GPT-4o, GeminiPro, Qwen2 and LLaMA-3.2 respectively. Within each partition, we first present results for direct inference with the MLLM, followed by ZS-CLIP, CoOp, and finally NeaR. Across all MLLMs, NeaR performs the best on average cACC, showing gains of at least +**3**% over the CoOp baseline for GPT-4o, GeminiPro and LLaMA-3.2, and a gain of +**1.3**% over CoOp for Qwen2. Furthermore, we observe a large performance gain for the difficult Car-196 dataset, where NeaR-LLaMA shows a gain of +**7.1**% in cACC over CoOp-LLaMA. These results highlight that NeaR effectively learns from the imperfect labels generated by MLLMs, leading to robust and efficient fine-grained classification.

**Comparison against PRODEN:** Our loss function resembles those used in Partial Label Learning (PLL), such as PRODEN (Feng et al., 2020), which are designed to handle learning with only candidate sets. To study the efficacy of traditional PLL approaches, we replace the traditional cross-entropy loss used in CoOp with PRODEN, and learn prompts using the candidate sets directly. The results are shown in Table 4, where NeaR outperforms PRODEN by a large margin of 4.3% in cACC and 3.2% in sACC. Unlike in traditional PLL where candidate sets are assumed to include the correct label, our candidate sets are generated for each image using noisy MLLM outputs and may not always contain the true label. Also, NeaR uniquely benefits from an initial "best-estimate" label $l_i$ from the MLLM, which traditional PLL methods do not exploit. As described in § 3.3, this best-estimate label is used to find "clean" images which have a higher probability of being correctly labeled. Knowledge from these clean samples helps resolve the ambiguity in candidate sets, improving performance.

**Comparison against JoAPR (Guo & Gu, 2024), a Contemporary Noisy Label Learning Method for CLIP:** JoAPR is a prompt-tuning method designed to fine-tune CLIP on noisy few-shot data. In Table 4, we show the results of using JoAPR to learn from noisy LLaMA generated labels. Our method NeaR outperforms JoAPR by +**4.6**% in average cACC, and by +**0.9**% in average sACC. For JoAPR we use the default configuration suggested in the paper. These gains highlight that generic noisy-label learning methods, which expect structured noise (such as flips) within a closed label set, do not fully address the challenges posed by open-ended MLLM outputs. By incorporating similarity information, performing candidate set guided label refinement, and performing label filtering, NeaR provides a robust solution to the VF-FGVR.

## 4.2 Ablation Studies

We conducted a thorough ablation to evaluate the contribution of each component in our pipeline in Table 5. We split the components of "Candidate Set Guided Label Refinement" into label mixup, rescaling, and using the candidate set guidance. The best performance is observed when all components — including warmup,

| Method | Bird-200 | | Car-196 | | Dog-120 | | Flower-102 | | Pet-37 | | Average | |
|---|---|---|---|---|---|---|---|---|---|---|---|---|
| | cACC | sACC | cACC | sACC | cACC | sACC | cACC | sACC | cACC | sACC | cACC | sACC |
| JoAPR-LLaMA | 49.2 | 70.0 | 42.8 | 60.6 | 59.5 | 70.6 | 76.7 | 60.1 | 73.9 | 83.3 | 60.4 | 68.9 |
| PRODEN | 48.3 | 67.6 | 45.9 | 60.6 | 57.9 | 67.0 | 75.2 | 59.0 | 75.8 | 78.5 | 60.6 | 66.6 |
| NeaR-LLaMA | 51.1 | 70.2 | 52.5 | 60.8 | 59.2 | 70.2 | 78.6 | 61.7 | 83.4 | 86.1 | **64.9 (+4.3%)** | **69.8 (+3.2%)** |

Table 4: Comparison of NeaR with a contemporary noisy label learning method, JoAPR (Guo & Gu, 2024), for CLIP using LLaMA-generated labels. NeaR outperforms JoAPR with an average improvement of +4.6% in cACC and +0.9% in sACC. These results indicate that directly applying LNL methods is insufficient to handle the challenges of noisy MLLM outputs. By incorporating better label refinement using candidate set, and by performing label filtering, NeaR provides a robust solution to the VF-FGVR problem. We also compare our method against PRODEN (Feng et al., 2020). We significantly outperform PRODEN on both cACC and sACC.

| Components | | | | | | Bird-200 | | Car-196 | | Dog-120 | | Flower-102 | | Pet-37 | | Avg | |
|---|---|---|---|---|---|---|---|---|---|---|---|---|---|---|---|---|---|
| W. | GMM | LM | R. | Cand. | LF | cACC | sACC | cACC | sACC | cACC | sACC | cACC | sACC | cACC | sACC | cACC | sACC |
| ✗ | ✓ | ✓ | ✓ | ✓ | ✓ | 48.8 | 71.5 | 49.8 | 61.9 | 56.8 | 71.9 | 75.4 | 61.3 | 79.3 | 85.4 | 62.0 | 70.4 |
| ✓ | ✗ | ✓ | ✓ | ✗ | ✓ | 48.5 | 69.1 | 48.9 | 59.7 | 57.1 | 69.6 | 76.9 | 60.1 | 81.8 | 84.1 | 62.6 | 68.5 |
| ✓ | ✓ | ✗ | ✓ | ✓ | ✓ | 51.3 | 71.3 | 48.6 | 60.7 | 59.4 | 71.4 | 75.9 | 60.1 | 79.7 | 84.0 | 63.0 | 69.5 |
| ✓ | ✓ | ✓ | ✗ | ✓ | ✓ | 49.1 | 69.4 | 50.9 | 60.6 | 59.5 | 71.8 | 77.1 | 61.4 | 77.8 | 84.1 | 62.9 | 69.5 |
| ✓ | ✓ | ✓ | ✓ | ✗ | ✗ | 48.0 | 69.1 | 48.3 | 60.2 | 56.6 | 71.0 | 73.9 | 60.2 | 80.6 | 84.4 | 61.5 | 69.0 |
| ✓ | ✓ | ✓ | ✓ | ✓ | ✗ | 49.1 | 70.2 | 50.1 | 60.1 | 57.9 | 70.1 | 75.6 | 60.6 | 81.4 | 85.9 | 62.8 | 69.3 |
| ✓ | ✓ | ✓ | ✓ | ✗ | ✓ | 48.5 | 69.1 | 48.9 | 59.7 | 57.1 | 69.6 | 76.9 | 60.1 | 81.8 | 84.1 | 62.6 | 68.5 |
| ✓ | ✓ | ✓ | ✓ | ✓ | ✓ | 51.0 | 70.2 | 52.6 | 60.9 | 59.2 | 70.2 | 78.6 | 61.7 | 83.4 | 86.1 | **65.0** | **69.8** |

Table 5: **Full ablation study across five datasets.** Columns "W." (Warm-up), "GMM", "LM" (Label Mixup), "R." (Rescaling), "Cand." (Candidate set), and "LF" (Label Filtering) indicate whether each module is enabled (✓) or disabled (✗). The best performance is achieved with all the components.

GMM-based partitioning, label mixup, rescaling, candidate set guidance, and post-training label filtering, are used. This configuration yields the highest average cACC (65.0%) and sACC (69.8%), re-confirming the effectiveness of proposed approach.

Removing the warm-up phase leads to a noticeable drop in performance, as it causes the GMM to produce a less reliable clean/noisy split. GMM is important because it splits the data into clean and noisily labeled samples. Without this split, the candidate set guidance is no longer used and only the MLLM generated labels are used throughout. Label mixup blends the label (one-hot or candidate set) with the model's prediction, and acts as a regularizer. Rescaling ensures that the refined label for a noisy sample has non-zero probabilities only for the candidate set, enabling it to learn from a relevant set of candidate labels. Removal of this leads to a misleading label signal for the noisy samples. Disabling either candidate set guidance or label filtering leads to noticeable drops across most datasets. To study the effect of removal of the candidate set guidance, we refine the labels of the noisy samples as $\bar{y}_i = \mathrm{shrp}(f_\theta(x_i), T)$, i.e we only used the sharpened CLIP pseudolabel. We also remove the candidate set based filtering $F_{cand}$, as defined in Sec 3.4. These ablations confirm that our components which are simple but carefully designed are integral for the performance gains.

**Imbalanced Training Data:** We study the realistic scenario of class imbalance in the few-shot training data. We simulate a long-tail distribution where we randomly select a small number of head classes (10 classes for pet-37) which have $4 \leq m \leq 10$ samples, and the remaining tail classes have $m = 3$ samples. We show results in Table 6. We observe that there is no degradation in performance for both CoOp and NeaR. Infact, we note slightly better cACC and sACC values for NeaR on account of the slight increase in the training data.

**Random Data Distribution** In this section we evaluate the performance of NeaR under extremely random data sampling. The results across all five datasets is shown in Table 7. Our method outperforms CoOp-LLaMA by 8.3% in cACC and 1.4% in sACC, demonstrating its robustness to data imbalance.

**Analysis on Number of Shots m in Training Data:** We explore the effect of the number of images used per class, as presented in Figure 2. We consistently use $\kappa = 3$ for candidate set construction across all shots.

| Method | Bird-200 | | Car-196 | | Dog-120 | | Flower-102 | | Pet-37 | | Average | |
|---|---|---|---|---|---|---|---|---|---|---|---|---|
| | cACC | sACC | cACC | sACC | cACC | sACC | cACC | sACC | cACC | sACC | cACC | sACC |
| FineR | 46.2 | 66.6 | 48.5 | 62.9 | 42.9 | 61.4 | 58.5 | 48.2 | 63.4 | 67.0 | 51.9 | 61.2 |
| ZS-CLIP-LLaMA | 48.9 | 67.0 | 46.9 | 60.3 | 55.9 | 64.5 | 71.4 | 58.5 | 75.5 | 72.2 | 59.7 | 64.5 |
| CoOp-LLaMA | 47.9 | 69.9 | 45.6 | 60.6 | 54.2 | 67.8 | 74.0 | 60.2 | 78.0 | 74.0 | 60.0 | 66.5 |
| NeaR-LLaMA | 50.9 | 69.9 | 52.6 | 60.4 | 60.2 | 71.2 | 80.3 | 63.8 | 84.6 | 86.2 | **65.7 (+5.7%)** | **70.3 (+3.8%)** |

Table 6: Performance comparison of NeaR-LLaMA with other baselines under long-tail class distribution. Both NeaR and CoOp retain performance on imbalanced data compared to balanced sampling. NeaR outperforms CoOp by +5.7% in cACC.

| Method | Bird-200 | | Car-196 | | Dog-120 | | Flower-102 | | Pet-37 | | Average | |
|---|---|---|---|---|---|---|---|---|---|---|---|---|
| | cACC | sACC | cACC | sACC | cACC | sACC | cACC | sACC | cACC | sACC | cACC | sACC |
| CoOp-LLaMA | 42.1 | 52.8 | 42.0 | 56.7 | 42.1 | 54.0 | 60.3 | 43.6 | 54.1 | 56.3 | 48.1 | 52.7 |
| NeaR-LLaMA | 43.0 | 52.0 | 47.8 | 55.3 | 55.8 | 55.7 | 73.5 | 52.0 | 61.9 | 55.7 | **56.4 (+8.3%)** | **54.1 (1.4%)** |

Table 7: Evaluation of NeaR under randomly sampled data distributions across five datasets. NeaR surpasses CoOp-LLaMA by 8.3% in cACC and 1.4% in sACC, highlighting its robustness to data imbalance.

For $m = 1$, our method performs poorly due to excessive label filtering. However, as the number of shots increases, our candidate set is more informative and performance improves markedly. Our proposed method, NeaR, outperforms CoOp for all $m \geq 2$, especially at higher shots where there are more noisy labels. We also observe that cACC drops with increasing $m$ due to increase in the size of the test time label space. Despite this, NeaR consistently outperforms others at all m. For example, at m=10, NeaR-LLaMA achieves 61.8 cACC, vs 54.7 (CoOp-LLaMA) and 52.2 (FineR). NeaR is more robust with increasing m with only a 3.5% drop from m=3 compared to 6% for others.

### 4.3 Impact of Varying No. of Nearest-Neighbors $\kappa$

We leverage similarity information to build a candidate set for each image by augmenting its label with the labels of its $\kappa$ nearest-neighbors. In this section we study the effect of varying $\kappa$ from 1 to 9 on the performance of NeaR-LLaMA. We perform this experiment for 9-shot data from the Flowers-102 dataset, to ensure that higher values of $\kappa$ give meaningful results. The results in Figure 3 show that NeaR performs well across a large range of $\kappa$ values, and justifies our choice of $\kappa = 3$. Note that setting $\kappa = 1$ is not the same as CoOp-LLaMA, but is the result of NeaR with $q_i = y_i$. The results also highlight two competing factors that influence the performance of NeaR as $\kappa$ varies:

- Improved Label Quality with Larger Candidate Sets – A larger candidate set is more likely to involve a semantically closer label. This is reflected in the upward trend of cACC from $\kappa = 1$ to $\kappa = 3$.

- Increased Noise with Larger Candidate Sets – For higher values of $\kappa$, while the likelihood of including better labels in the candidate set increases, it is offset by the addition of irrelevant labels. A noisier candidate set makes it harder for the algorithm to disambiguate the best label in the candidate set. This leads to a plateau or even slight decrease in cACC for $\kappa > 3$.

To generalize this idea, we conduct an experiment where m is randomly sampled between 1 and 10 as shown in Table 8, and we ablate over different values of k ranging from 1 to 9. We find that the cACC remains relatively stable across this range, with only minor variations. This indicates that our method is not particularly sensitive to the choice of k, which can be selected flexibly. In our implementation, we use k=3 as a practical default.

### 4.4 Choice of Vision Encoder for Candidate Set Construction

In this section we study how alternative choices of image embeddings for selecting top-$\kappa$ nearest neighbors affect the performance of NeaR. We use the DINO (Caron et al., 2021) and MAE (He et al., 2021) encoders

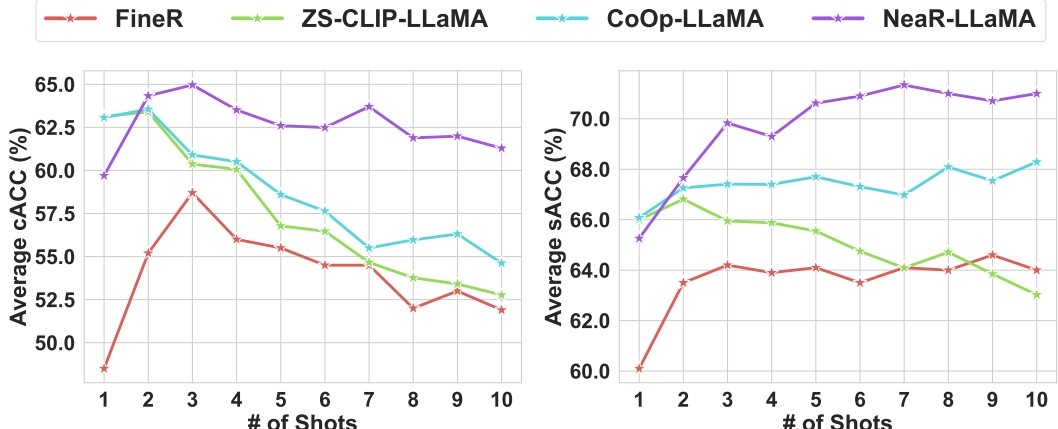

Figure 2: Effect of varying $m$, number of images per class in training data for labels generated by LLaMA. NeaR (in purple) outperforms CoOp (blue), ZSCLIP-LLaMA (green) and FineR (red) for all $m \geq 2$ in average cACC & sACC, where there are more noisy labels. For $m=1$, our method performs poorly due to excessive label filtering.

| Method ($k$-nearest) | cACC | sACC |
|:---:|:---:|:---:|
| $k = 1$ | 69.8 | 50.1 |
| $k = 2$ | 72.3 | 51.7 |
| $k = 3$ | 73.5 | 52.0 |
| $k = 4$ | 74.7 | 53.9 |
| $k = 5$ | 74.0 | 53.1 |
| $k = 6$ | 74.3 | 53.6 |
| $k = 7$ | 73.6 | 53.8 |
| $k = 8$ | 71.5 | 52.6 |
| $k = 9$ | 70.9 | 52.0 |

Table 8: To study the generalized idea of effect of the candidate-set size ($k$) on NeaR, we perform an experiment where m is randomly sampled between 1 and 10 for Flower-102.

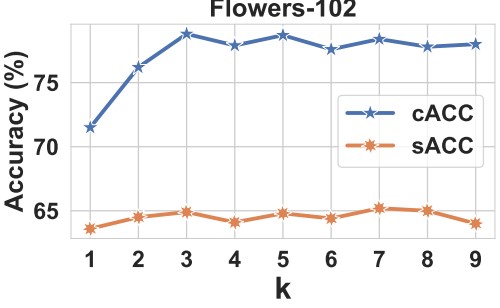

Figure 3: Effect of varying $\kappa$ (1 to 9) on the performance of NeaR-LLaMA for the 9-shot Flowers-102 dataset. Note that setting $\kappa = 1$ is not the same as CoOp-LLaMA, but is the result of NeaR with $q_i = y_i$.

solely for the nearest neighbor candidate set construction step, while retaining our default ViT-B/16 CLIP backbone for prompt tuning. The results for Llama-3.2, presented in Table 9, show that our method continues to achieve competitive performance even when the k-NN step is based on features from other backbones.

| Image Encoder | Bird-200 | | Car-196 | | Dog-120 | | Flower-102 | | Pet-37 | | Average | |
|:---|:---:|:---:|:---:|:---:|:---:|:---:|:---:|:---:|:---:|:---:|:---:|:---:|
| | cACC | sACC | cACC | sACC | cACC | sACC | cACC | sACC | cACC | sACC | cACC | sACC |
| DINO | 50.3 | 70.5 | 50.6 | 61.8 | 60.4 | 72.6 | 79.7 | 60.8 | 82.4 | 88.3 | 64.7 | 70.8 |
| MAE | 48.6 | 70.1 | 50.9 | 61.5 | 59.9 | 72.2 | 76.6 | 59.8 | 84.8 | 88.6 | 64.2 | 70.4 |
| CLIP ViT-B/16 | 51.03 | 70.2 | 52.6 | 60.9 | 59.2 | 70.2 | 78.6 | 61.7 | 83.4 | 86.1 | 65.0 | 69.8 |

Table 9: Evaluation of NeaR-LLaMA when different image encoders are used to perform nearest neighbor selection during candidate set construction. The default CLIP ViT-B/16 backbone is retained for the rest of the pipeline. We observe that performance remains invariant to the choice of image embeddings chosen to construct the candidate sets.

| Method | Bird-200 | | Car-196 | | Dog-120 | | Flower-102 | | Pet-37 | | Average | |
|---|---|---|---|---|---|---|---|---|---|---|---|---|
| | cACC | sACC | cACC | sACC | cACC | sACC | cACC | sACC | cACC | sACC | cACC | sACC |
| RN50 | | | | | | | | | | | | |
| CoOp-LLaMA | 13.9 | 44.1 | 8.6 | 46.6 | 18.2 | 47.9 | 15.6 | 31.7 | 36.7 | 54.7 | 18.6 | 45.0 |
| NeaR-LLaMA | 17.3 | 50.5 | 9.8 | 48.9 | 19.9 | 49.9 | 20.5 | 36.6 | 46.1 | 63.3 | **22.7 (+4.1%)** | **49.8 (+4.8%)** |
| RN101 | | | | | | | | | | | | |
| CoOp-LLaMA | 17.3 | 45.5 | 9.6 | 47.4 | 20.8 | 45.4 | 17.5 | 34.7 | 44.0 | 55.2 | 21.8 | 45.6 |
| NeaR-LLaMA | 18.6 | 49.6 | 11.4 | 50.8 | 22.4 | 51.8 | 18.9 | 36.1 | 47.0 | 63.9 | **23.6 (+1.8%)** | **50.5 (+4.9%)** |
| ViT-B/16 | | | | | | | | | | | | |
| CoOp-LLaMA | 49.2 | 68.7 | 45.5 | 60.7 | 58.4 | 68.4 | 75.9 | 59.8 | 74.4 | 79.2 | 60.7 | 67.4 |
| NeaR-LLaMA | 51.0 | 70.2 | 52.6 | 60.9 | 59.2 | 70.2 | 78.6 | 61.7 | 83.5 | 86.2 | **65.0 (+4.3%)** | **69.8 (+2.4%)** |
| ViT-B/32 | | | | | | | | | | | | |
| CoOp-LLaMA | 45.0 | 56.3 | 39.3 | 60.5 | 51.9 | 66.2 | 69.8 | 59.1 | 72.7 | 79.8 | 55.7 | 66.2 |
| NeaR-LLaMA | 48.8 | 68.4 | 47.8 | 60.0 | 56.4 | 68.9 | 75.0 | 61.3 | 77.3 | 82.0 | **61.1 (+5.4%)** | **68.1 (+1.9%)** |
| ViT-L | | | | | | | | | | | | |
| CoOp-LLaMA | 55.8 | 73.4 | 51.1 | 63.0 | 65.1 | 74.3 | 81.7 | 63.3 | 83.2 | 87.6 | 67.3 | 72.3 |
| NeaR-LLaMA | 54.5 | 73.1 | 62.5 | 62.3 | 67.9 | 74.6 | 82.4 | 64.7 | 84.9 | 87.3 | **70.5 (+3.2%)** | **72.4 (+0.1%)** |
| ViT-G | | | | | | | | | | | | |
| CoOp-LLaMA | 56.2 | 74.0 | 51.2 | 63.1 | 64.7 | 73.8 | 82.2 | 63.1 | 82.4 | 87.5 | 67.3 | **72.3** |
| NeaR-LLaMA | 59.0 | 74.9 | 63.0 | 61.7 | 66.2 | 74.2 | 87.5 | 64.9 | 83.7 | 85.0 | **71.9 (+4.6%)** | 72.1 (-0.2%) |
| SigLIP | | | | | | | | | | | | |
| CoOp-LLaMA | 36.8 | 64.1 | 61.9 | 62.8 | 58.5 | 70.2 | 71.1 | 57.2 | 75.6 | 80.0 | 60.8 | **66.9** |
| NeaR-LLaMA | 34.0 | 63.0 | 71.5 | 62.9 | 57.7 | 70.2 | 70.9 | 56.7 | 72.9 | 78.6 | **61.4 (+0.6%)** | 66.3 (-0.6%) |

Table 10: Performance comparison of NeaR-LLaMA with CoOp-LLaMA across different CLIP backbones, including ResNet-50 (RN50), ResNet-101 (RN101), ViT-B/32, ViT-L, ViT-G and a different pretraining startegy SigLIP. For completeness, results are also provided for the default backbone, ViT-B/16. NeaR consistently outperforms CoOp-LLaMA, achieving gains of +4.1% and +4.8% in cACC and sACC for RN50, +1.8% and +4.9% in cACC and sACC for RN101, and +5.4% and +1.9% in cACC and sACC for ViT-B/32, +3.2% in cACC and +0.1% in sACC for ViT-L, +4.6% in cACC for ViT-G and +0.6% in cACC for SigLIP. NeaR consistently outperforms the baseline for CLIP models.

## 4.5 Performance of NeaR on Different CLIP Backbones:

All results in this paper are on the ViT-B/16 CLIP backbone. In this section we compare the performance of NeaR-LLaMA with CoOp-LLaMA across various other CLIP backbones and SigLIP. In Table 10 we present the results of NeaR for a ResNet-50 (He et al., 2015), ResNet-101, and a ViT-B/32 vision-encoder based CLIP model. We use the same configuration for each backbone. We observe performance gains of 5.5% in cACC and 3.2% in sACC with ViT-L, and gains of 6.9% in cACC and 2.9% in sACC with ViT-G over the default CLIP ViT-B/16 backbone. Our method outperforms CoOp-LLaMA by 3.2% for ViT-L and by 4.6% for ViT-G, demonstrating its effectiveness across stronger backbone architectures. We also experiment with the SigLIP ViT-B/16 backbone which has a different pretraining objective compared to CLIP. Our results indicate that our method does not transfer effectively to SigLIP. We conjecture that our method fails because our final objective resembles the InfoNCE objective used by CLIP, and is different from the Sigmoidal loss employed by SigLIP. However, we are competitive with the CoOp baseline in avg cACC. These results highlight the effectiveness of our method over a diverse range of CLIP architectures.

## 4.6 Study on Adapter based Finetuning

In this section we follow CLIP-Adapter (Gao et al., 2021) to finetune linear adapters added on top of both frozen CLIP image and text encoders, as an alternative to prompt-tuning. We begin by noting that prompt learning is more efficient in terms of parameter count. In our setup, we use 16 prompt tokens for a total of $16 * 512$ learnable parameters, whereas a fully-connected linear layer has $512 * K$ parameters where K is the number of labels/classes (eg. number of GPT-4o generated labels is K=300 for Bird-200). The label filtering step § 3.4 in our formulation results in a variable label space during test time, and thus we cannot directly

train a linear probe with a fixed number of classes. The table below shows cACC results for $\alpha = 0.8$ (from (Gao et al., 2021)) for GPT-4o generated labels. The first row simply shows the naive implementation of the CLIP Adapter (Gao et al., 2021) on the labels. For the next two rows, we observe that NeaR-Adapter-GPT4o outperforms the adapter baseline (59.98% vs 48.75% average cACC), but still falls short of the performance achieved by prompt-based NeaR-GPT4o (59.98% vs 67.6% average cACC). This suggests that adapter based tuning is unsuitable for the VF-FGVR setting, potentially due to its reduced robustness to open-vocabulary label noise. This is consistent with findings from (Wu et al., 2023) which shows that prompt tuning is more robust to label noise compared to linear classifiers.

| Method | Bird-200 | | Car-196 | | Dog-120 | | Flower-102 | | Pet-37 | | Average | |
|---|---|---|---|---|---|---|---|---|---|---|---|---|
| | cACC | sACC | cACC | sACC | cACC | sACC | cACC | sACC | cACC | sACC | cACC | sACC |
| Adapter-GPT4o | 39.2 | 63.9 | 40.4 | 56.4 | 40.0 | 61.1 | 57.7 | 42.9 | 66.2 | 75.8 | 48.7 | 60.0 |
| NeaR-Adapter-GPT4o | 50.2 | 73.9 | 48.0 | 59.2 | 54.7 | 69.7 | 70.2 | 49.3 | 76.6 | 82.3 | 59.9 | 66.9 |
| NeaR-GPT4o (Ours) | 55.8 | 75.6 | 57.0 | 60.0 | 61.6 | 74.4 | 80.6 | 52.1 | 82.9 | 84.0 | **67.6 (+7.7%)** | **69.2 (+2.3%)** |

Table 11: Comparison of prompt-based NeaR-GPT4o (Ours) with adapter based variant NeaR-Adapter-GPT4o and standard adapter finetuning.

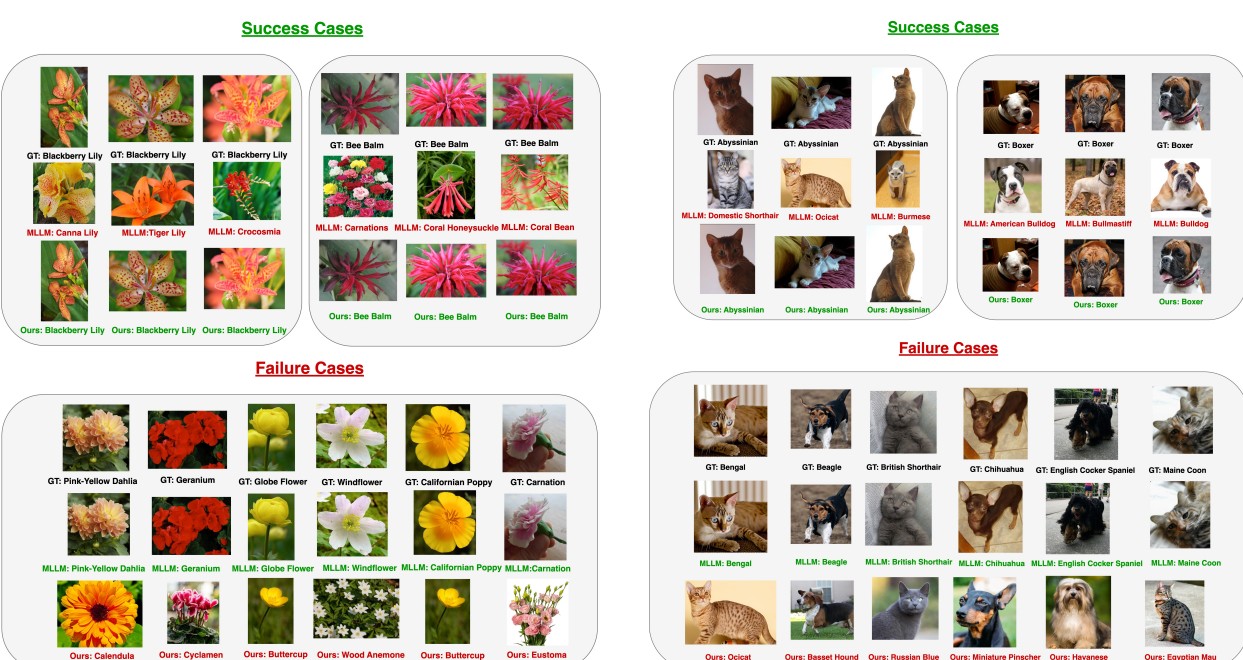

Figure 4: Qualitative results showcasing both success and failure cases of Flower-102 dataset (left) and Pet-37 dataset (right). Success refers to instances where NeaR correctly predicts the class label while the MLLM fails. Failure cases illustrate examples where NeaR produces incorrect predictions.

## 4.7 Qualitative Results of NeaR.

We visualize a selection of inference images and analyze their predictions for the Flower-102 and Pet-37 datasets in Figure 4. Specifically, we compare the predictions of NeaR with those obtained from directly querying the MLLM. When comparing sACC, we observe that MLLMs (GPT4/GeminiPro/LLaMA) produce stronger labels (sACC:72.4/73.3/69.9) or predictions than FineR (sACC: 64.3). Therefore, we focus our comparisons on NeaR against MLLMs. In success cases (e.g., Blackberry Lily, Bee Balm), NeaR predicts correct labels consistently, while LLaMA produces inconsistent and unrelated labels like Carnation or Coral Honeysuckle, reflected in lower cACC. In failure cases, NeaR often misclassifies flowers as other species with similar structures or colors (e.g., Globe Flower → Buttercup, Windflower → Wood Anemone), unlike LLaMA, which shows greater label spread. For classes that NeaR underperforms, we find they are harder overall, even

for the MLLM. Even when incorrect, NeaR is more consistent (e.g., Windflower $\rightarrow$ Wood Anemone in most test images), while LLaMA outputs 7 diverse labels for same set of images. We observe a similar trend for pets.

## 5   Conclusion

We addressed the challenge of Vocabulary-Free Fine-Grained Visual Recognition (VF-FGVR) by introducing NeaR, a method that leverages MLLMs to generate weakly supervised labels for a small set of training images, to efficiently fine-tune a downstream CLIP model. Our approach constructs a candidate label set for an image using generated labels of similar images, and performs label refinement for clean and noisy data differently. NeaR also proposes a label filtering strategy, effectively managing the open-ended and noisy nature of MLLM outputs. Experiments on 4 MLLMs show that NeaR significantly outperforms direct inference methods while dramatically reducing computational cost and inference time, setting a new benchmark for efficient and scalable VF-FGVR.

## Impact Statement

Our framework relies on MLLM-generated labels, a dependency that is becoming increasingly feasible with advancements in MLLM accessibility. We demonstrate strong performance across both proprietary (GPT-4o, GeminiPro) and open-source (LLaMA-11B, Qwen2-7B) models, showing robustness to MLLMs of varying capacities. We see this work as a foundation for future research in leveraging MLLMs for fine-grained recognition, with no direct societal or ethical risks.

## Acknowledgements

Hari Chandana Kuchibhotla, Sai Srinivas Kancheti and Abbavaram Gowtham Reddy would like to thank MoE for the generous PMRF fellowship support to each of them; Sai Srinivas Kancheti and Vineeth N Balasubramanian would like to thank Microsoft Research India for the MSR India PhD Award and the AFMR grant, under which this work was carried out. We thank the anonymous reviewers for their valuable feedback that improved the presentation of this paper.

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

## Appendix

This supplementary material contains additional details:

- Summary of notations and their descriptions in § A6

- The overall NeaR algorithm is presented in § A7

- Further Comments on Efficiency of NeaR in § A8.1.

- Further descriptions of datasets used in § A8.2.

- Effect of label filtering on the size of label space in § A8.3

- Analysis on an alternative way to obtain candidate sets is shown in § A8.4

- Performance across different prompting strategies in shown in § A8.5

- The effect of choice of threshold $\tau$ is studied in § A8.6

- Comparison with clustering-based methods in § A8.7

- Performance across variations in different few-shot splits is shown in § A8.8

- Comparison with other Pseudo-labeling based Methods in § A8.9

- Analysis of retrieved images in § A8.10

- Visualizing noisy samples from GMM in § A8.11

- Discussion on other weaker open-source MLLMs is presented in § A8.12

- Prompts used to generate labels for different MLLMs in § A8.13

## A6    Summary of Notations.

We represent elements of a set by a subscript and a vector component by a superscript. For instance $y_i \in \{0,1\}^k$ denotes the one-hot vector encoding of the class label of the $i$-th image, $i \in [n]$, and $y_i^j$ is the $j$-th component of this encoding. Specifically $y_i^j = 1$ if the $i$-th image is assigned the $j$-th label in the label set $\mathcal{C}$ and 0 otherwise. A summary of notations is given in Table A12.

## A7    NeaR Algorithm

We present the training algorithm of NeaR in Algorithm 1. We present the three steps of our method NeaR as shown in Figure 1. In lines L1-L7 we generate possibly noisy labels from an MLLM. In lines L8-L12, we then generate a candidate set from $\kappa$ nearest-neighbors of each image. In lines L19-L27 we train prompt vectors $\theta$ by minimizing the cross-entropy loss between CLIP predictions and the refined labels. The label refinement in L24 effectively disambiguates the best label from the generated candidates. As training progresses, our estimate of the candidate labels $q_i$ gets better. The sharpening function is $\mathrm{shrp}(y,T)^i = (y^i)^{\frac{1}{T}} / \sum_{j=1}^{k} (y^j)^{\frac{1}{T}}$ and the rescale function is defined as $\mathrm{rsc}(y,q)^i = (y \odot q)^i / \sum_{j=1}^{k} (y \odot q)^j$.

| Notation | Description |
|---|---|
| $x_i$ | $i$-th unlabeled training image |
| $X$ | set of $n$ training images $\{x_1, x_2, \ldots, x_n\}$ |
| $m$ | Number of shots of images for each class belonging to an unknown ground-truth class name set |
| $l_i = L(x_i, p)$ | The class label generated for $x_i$ by an MLLM $L$ with a input prompt $p$ |
| $\mathcal{D} = \{(x_i, l_i)\}_{i=1}^n$ | The dataset generated by an MLLM $L$ consisting of image-class name pairs |
| $\mathcal{C} = \bigcup_{i=1}^n l_i$ | Label space generated by the MLLM |
| $k = |\mathcal{C}|$ | The $k$ lexicographically ordered class names $\mathcal{C} = \{c_1, c_2, \ldots, c_k\}$ |
| $y_i \in \{0, 1\}^k$ | One-hot encoding of the label $l_i$ of $x_i$ |
| $\mathcal{I}$ | Image encoder of pre-trained CLIP |
| $\mathcal{T}$ | Text encoder of pre-trained CLIP |
| $\theta$ | Learnable prompt vectors added to the input embeddings of a class name |
| $f_\theta(x) \in \Delta^k$ | $k$-dimensional probability vector of the prompted CLIP model's class predictions for image $x$, i.e $f_\theta(x)^j \geq 0$ and $\sum_{j=1}^k f_\theta(x)^j = 1$ |
| $S_i$ | Candidate set created by gathering MLLM generated labels of nearest-neighbors of $x_i$ |
| $\kappa$ | Number of nearest-neighbors |
| $\mathcal{D}_s = \{(x_i, l_i, S_i)\}_{i=1}^n$ | Augmented dataset containing the generated label $l_i$ and constructed candidate set $S_i$ |
| $L_{ce}(f_\theta(x_i), l_i)$ | The cross-entropy loss of CLIP model predictions for image $i$ w.r.t $y_i$ given by $-\sum_{j=1}^k y_i^j log(f_\theta^j(x_i))$ (also written as $L(x_i, l_i)$) |
| $GMM$ | A two-component Gaussian Mixture Model fit on loss values of all training samples for every epoch |
| $w_i = \mathbb{P}_{GMM}(clean|x_i)$ | The posterior probability of image belonging to the "clean" component, i.e component with lower mean |
| $\tau \in [0, 1]$ | Threshold used to partition data into clean and noisy sets |
| $X_{cl}, X_{ns}$ | Clean and noisy partitions of the training data based on $w_{\geq \tau}$ |
| $q_i \in \mathbb{R}^k$ | Confidence of the candidates. We have that $q_i^j > 0$ if $c_j \in S_i$, $q_i^j = 0$ otherwise, and $\sum_{j=1}^k q_i^j = 1$ |
| $\bar{y}_i$ | Refined label for image $i$ constructed based on whether $x_i$ is clean or noisy. For noisy images we rescale the label to have non-zero probabilities only for the candidate labels |
| $\mathrm{shrp}(y, T)$ and $\mathrm{rsc}(y, q)$ | sharpen a distribution $y$ using temperature $T$; rescale a distribution $y$ based on candidate confidence $q$ |
| $L_{final}(\theta)$ | The cross entropy loss between model predictions $f_\theta(x)$ and refined labels $\bar{y}$ |
| $\mathcal{C}_{test}$ | Inference label space post filtering |

Table A12: List of notations used in our paper, and their descriptions.

---

**Algorithm 1** NeaR algorithm: Training

---

**Require:** $m$-shot training images $X = \{x_i\}_{i=1}^n$; MLLM $L$; input prompt $p$; number of nearest neighbors $\kappa$; CLIP model predictions $f_\theta$ with learnable text prompts $\theta$; $num\_epochs$; $warm\_epochs$; learning-rates $\eta$, $\eta_{warm}$; Temperature $T$

**Ensure:** Trained parameters $\hat\theta$

  **Step-1: Labeling training images with an MLLM**

1: $D \leftarrow \{\}$

    */* Label each image $x$ by prompting MLLM $L$ with prompt $p$ */*

2: **for** $i = 1, 2, \ldots, n$ **do**

3:     $D \leftarrow D \cup \{(x_i, l_i := L(x_i, p))\}$

4: **end for**

5: $\mathcal{C} := \bigcup_{i=1}^n l_i$

6: $k := |\mathcal{C}|$

    */* WLOG we consider $\mathcal{C} := \{c_1, c_2, \ldots, c_k\}$ where $k := |\mathcal{C}|$ to be lexicographically ordered. Let $y_i$ be the one-hot encoding of label $l_i$ */*

7: $y_i \in \{0, 1\}^k$ and $y_i^j := 1$ if $c_j = l_i$ and 0 o/w

  **Step-2: Candidate Set Construction**

8: $D_s \leftarrow \{\}$

    */* Augment each image $x_i$ with a candidate set $S_i$ composed of labels of $\kappa$-nearest neighbors */*

9: **for** $(x_i, l_i)$ in $D$ **do**

10:     $S_i \leftarrow$ knn_labels$(x_i, \kappa)$

11:     $D_s \leftarrow D_s \cup \{(x_i, l_i, S_i)\}$

12: **end for**

    */* We initialize candidate confidence for all images $q_i$ uniformly */*

13: $q_i \in \mathbb{R}^k$ and $q_i^j := \frac{1}{|S_i|}$ if $c_j \in S_i$ and 0 o/w

  Function Partition_data($D$, $f_\theta$, $\tau$):

14:     $\mathcal{L} := \{L(f_\theta(x_i), l_i)\}_{i=1}^n$

15:     $\mu_c, \sigma_c, \mu_n, \sigma_n \leftarrow fit\_GMM(\mathcal{L})$

16:     $W := \{w_1, w_2, \ldots, w_n\}$ where $w_i = \mathbb{P}_{GMM}(clean|x_i)$

17:     $X_{cl} := \{x_i \in X \mid w_i \geq \tau\}$

18:     $X_{ns} := X \setminus X_{cl}$

  return $X_{cl}, X_{ns}, W$

  **Step-3: Fine-tune prompts $\theta$ of a CLIP model**

19: **for** $t = 1, 2, \ldots, num\_epochs$ **do**

    */* During warmup, the prompts are tuned on the cross-entropy loss $L_{ce}$ using MLLM generated labels in $D$ */*

20:     **if** $t \leq warm\_epochs$ **then**

21:       $\theta_t \leftarrow \theta_{t-1} - \eta_{warm} \nabla L_{ce}(D, f_{\theta_{t-1}})$

22:     **else**

    */* Every epoch post warm-up, we partition the entire data into clean and noisy sets by fitting a GMM on cross-entropy loss $L$ */*

23:       $X_{cl}, X_{ns}, W \leftarrow$ Partition_data$(D, f_{\theta_{t-1}}, \tau)$

24:       $\bar{y}_i := \text{shrp}\left(w_i \cdot y_i + (1 - w_i) \cdot f_{\theta_{t-1}}(x_i), T\right), \quad \text{if } x_i \in X_{cl}$

          $:= \text{rsc}\left(\text{shrp}(w_i \cdot q_i + (1 - w_i) \cdot f_{\theta_{t-1}}(x_i), T), q_i\right), \text{ o/w}$

    */* We update candidate confidence (for both clean and noisy samples) to be used in the next epoch. $\mathbb{1}[q_i]$ is 1 at non-zero indices and 0 o/w */*

25:       $q_i \leftarrow \text{rsc}(f_{\theta_{t-1}}(x_i), \mathbb{1}[q_i])$

26:       $L_{final}(\theta_{t-1}) := \frac{-1}{n} \sum_{i=1}^n \sum_{j=1}^k \bar{y}_i^j \cdot log(f_{\theta_{t-1}}^j(x_i))$

27:       $\theta_t \leftarrow \theta_{t-1} - \eta \nabla L_{final}(\theta_{t-1})$

28:     **end if**

29: **end for**

30: **Return:** $\hat\theta = \theta_{num_{epochs}}$

---

## A8    Additional Results

In this section we start with further comments on Efficiency of NeaR in § A8.1. Descriptions of the datasets in A8.2 used followed by additional results. i) Effect of label filtering on the size of label space in § A8.3. ii) Analysis on an alternative way to obtain candidate sets is shown in § A8.4. iii) Performance across different prompting strategies in shown in § A8.5. iv) The effect of choice of threshold $\tau$ is studied in § A8.6. v) Comparison with clustering-based methods in § A8.7. vi) Performance across variations in different few-shot splits is shown in § A8.8. vii) Comparison with other Pseudo-labeling based Methods in § A8.9. viii) Analysis of retrieved images in § A8.10. ix) Visualizing noisy samples from GMM in § A8.11. x) Discussion on other weaker open-source MLLMs is presented in § A8.12. xi) Prompts used to generate labels for different MLLMs in § A8.13.

### A8.1    Further Comments on Efficiency of NeaR

Table 1 comprehensively covers the different MLLM types and all baselines we designed. For instance, we report inference time, cost, and cACC for GPT-4o, which closely reflects performance for Gemini Pro as well. Similarly, results for LLaMA are representative of Qwen. For ZS-CLIP-MLLM, CoOp-MLLM, and NeaR-MLLM, inference time and cost remain the same, with only a slight overhead in training time. We also note that the cost estimates presented were for GPT-4o in late 2024, and current API costs may be cheaper as inference becomes more scalable. We acknowledge that our cost estimate ( \$100 for 32k images) could be improved by leveraging batched inference APIs, which can indeed reduce the cost by up to 5× and ensure timely completion. Our original estimate was based on standard per-image API usage (e.g. GPT-4 or Gemini Pro), which many users adopt by default due to ease of integration. However, we agree that using optimized batch pipelines is a more efficient option, potentially reducing the cost to approximately \$20 for 32k images. Even by using optimized batched pipeline, our method always requires lesser costs.

### A8.2    Description of Datasets Used.

We show results on 5 datasets with fine-grained labels – Bird-200, Car-196, Dog-120, Flower-102, Pet-37. In Table A13, we show the number of images used for training and the size of the test set.

|  | **Bird-200** | **Car-196** | **Dog-120** | **Flower-102** | **Pet-37** |
|---|---|---|---|---|---|
| Train Set | $m \times 200$ | $m \times 196$ | $m \times 120$ | $m \times 102$ | $m \times 37$ |
| Test Set | 5794 | 8041 | 8550 | 6149 | 3669 |

Table A13: Train and test set sizes of the datasets used in this paper. The number of shots is denoted by $m$, with $m = 3$ used as the default in our experiments unless otherwise specified.

### A8.3    Effect of Label Filtering on the Size of Label Space.

As described in § 4.2, label filtering is crucial to obtain good VF-FGVR performance. In Table A14, we present the number of classes in the final classification label spaces that each method operates in. The first row indicates the size of the ground-truth label space. We observe that our label filtering mechanism is essential to combat the open-endedness of MLLM labels.

| Method | Average | | | | |
|---|---|---|---|---|---|
|  | Bird-200 | Car-196 | Dog-120 | Flower-102 | Pet-37 |
| Ground Truths | 200 | 196 | 120 | 102 | 37 |
| MLLM Labels | 412 | 562 | 169 | 183 | 63 |
| FineR | 202 | 286 | 97 | 112 | 44 |
| NeaR-LLaMA | 239 | 305 | 129 | 119 | 45 |

Table A14: Label filtering is effective in reducing the size of MLLM generated label to manageable levels.

### A8.4 Analysis on Alternative Ways to Construct Candidate Sets

The labels generated by MLLMs can be noisy. To address this, we propose to construct a candidate set for each image by grouping class labels from the $\kappa$ nearest-neighbors of the image. In this section we study an alternative approach to candidate set generation, where we query the MLLM itself to generate a set of $\kappa$ labels directly for each image. We present the results of this approach in Table A15, showing performance across three MLLMs: GPT-4o, GeminiPro, and LLaMA. The purpose of this experiment is to demonstrate the effectiveness of candidate set construction via KNN. For the Car-196 dataset, the generated candidate labels were highly diverse—resulting in a union of approximately 1200 unique labels across all images which leads to OOM issues during text feature computation. As a workaround for the Car-196 dataset alone, we limited the candidate sets to the top-2 labels per image. Furthermore, for LLaMA, our $\kappa$-nn based approach outperforms the direct approach by a substantial margin, achieving a **+5.2**% higher cACC, while being more efficient.

| Method | Bird-200 | | Car-196 | | Dog-120 | | Flower-102 | | Pet-37 | | Average | |
|---|---|---|---|---|---|---|---|---|---|---|---|---|
| | cACC | sACC | cACC | sACC | cACC | sACC | cACC | sACC | cACC | sACC | cACC | sACC |
| NeaR-GPT-4o-Direct | 53.1 | 74.2 | 46.0 | 54.1 | 61.0 | 71.9 | 77.0 | 55.4 | 83.1 | 82.7 | 64.0 | 67.6 |
| NeaR-GPT-4o | 55.8 | 75.6 | 57.0 | 60.0 | 61.6 | 74.4 | 80.6 | 52.1 | 82.9 | 84.0 | **67.6 (+3.6%)** | **69.2 (+1.6%)** |
| NeaR-GeminiPro-Direct | 52.7 | 73.8 | 43.8 | 52.2 | 70.5 | 58.3 | 73.9 | 46.9 | 81.3 | 81.7 | 64.4 | 62.5 |
| NeaR-GeminiPro | 55.9 | 76.0 | 54.9 | 61.1 | 64.7 | 75.4 | 77.9 | 53.2 | 79.4 | 80.8 | **66.6 (+2.2%)** | **69.3 (+6.8)** |
| NeaR-LLaMA-Direct | 43.8 | 68.2 | 44.7 | 56.0 | 56.4 | 70.6 | 70.9 | 57.7 | 83.2 | 87.5 | 59.8 | 68.0 |
| NeaR-LLaMA | 51.0 | 70.2 | 52.6 | 60.9 | 59.2 | 70.2 | 78.6 | 61.7 | 83.5 | 86.2 | **65.0 (+5.2%)** | **69.8 (+1.8%)** |

Table A15: Evaluation of NeaR-MLLM under different candidate set generation methods. We compare our $\kappa$-nn-based candidate set against directly querying the MLLM for a candidate set, referred to as NeaR-MLLM-Direct. For the Car-196 dataset, both GPT-4o and GeminiPro encounter Out-of-Memory (OOM) errors due to the larger label space. For NeaR-LLaMA, our $\kappa$-nn-based approach outperforms the direct approach by an average margin of +5.2% in cACC while being more computationally efficient.

### A8.5 Different Prompting Strategies.

We analyze the impact of our proposed approach using various prompting strategies, as presented in Table A16. We consider three distinct prompting methods involving fine-tuning across different modalities: (1) For text-only prompting, we use CoOp (Zhou et al., 2021); (2) For image-only prompting, we employ VPT (Jia et al., 2022); and (3) For both text and image prompting, we adopt hierarchical prompts introduced at different text and image layers (Rasheed et al., 2022) as the backbone. Our method demonstrates strong performance across all prompting strategies, achieving cACC improvements of 4.2%, 1.9%, and 4.1%, respectively. This clearly demonstrates the effectiveness of our method across different fine-tuning methods.

| Method | Bird-200 | | Car-196 | | Dog-120 | | Flower-102 | | Pet-37 | | Average | |
|---|---|---|---|---|---|---|---|---|---|---|---|---|
| | cACC | sACC | cACC | sACC | cACC | sACC | cACC | sACC | cACC | sACC | cACC | sACC |
| Text Prompting | | | | | | | | | | | | |
| CoOp-LLaMA | 49.2 | 68.7 | 45.5 | 60.7 | 58.4 | 68.4 | 75.9 | 59.8 | 74.4 | 79.2 | 60.7 | 67.4 |
| NeaR-LLaMA | 51.0 | 70.2 | 52.6 | 60.9 | 59.2 | 70.2 | 78.6 | 61.7 | 83.5 | 86.2 | **65.0 (+4.3%)** | **69.8 (+2.4%)** |
| Visual Prompting | | | | | | | | | | | | |
| VPT-LLaMA | 48.9 | 69.5 | 45.3 | 61.3 | 60.3 | 70.2 | 78.1 | 61.7 | 73.2 | 82.2 | 61.1 | 69.0 |
| NeaR-VPT-LLaMA | 50.2 | 68.5 | 45.5 | 60.1 | 59.4 | 71.4 | 78.3 | 61.3 | 81.1 | 84.3 | **62.9 (+1.8%)** | **69.1 (+0.1%)** |
| Multimodal Prompting | | | | | | | | | | | | |
| IVLP-LLaMA | 48.9 | 69.5 | 45.3 | 61.3 | 60.3 | 70.2 | 78.1 | 61.7 | 73.2 | 82.2 | 61.1 | 69.0 |
| NeaR-IVLP-LLaMA | 50.8 | 70.1 | 52.5 | 61.0 | 58.6 | 69.9 | 80.3 | 62.1 | 83.7 | 86.4 | **65.2 (+4.1%)** | **69.9 (+0.9%)** |

Table A16: Evaluation of NeaR under different prompting strategies. In addition to text-based prompting, as shown in Table 3, we present results on Visual Prompting method VPT (Jia et al., 2022) and Multimodal Prompting method IVLP (Rasheed et al., 2022). We outperform the baselines by +1.8% and +4.1% in cACC respectively.

### A8.6 Effect of Choice of Threshold $\tau$

To address the noisy nature of MLLM generated labels, our method NeaR separates samples into clean and noisy sets using a threshold $\tau$ based on clean posterior probability $w_i$ of a GMM fitted on loss values. Instead of using a fixed threshold, we make $\tau$ adaptive by setting it to the mean posterior probability, $\tau = \frac{1}{n} \sum_{i=1}^{n} w_i$, allowing dynamic estimation of label noise at every training epoch. We study the effects of using a fixed threshold of $\tau = 0.5$ for NeaR-LLaMA, NeaR-GPT-4o, NeaR-GeminiPro and NeaR-Qwen2 in Table A17. We observe that for NeaR-GeminiPro and NeaR-Qwen2, we have a performance gain of $+1.3\%$ and $+1.1\%$ in average cACC, while a relatively lower performance gain of $+0.2\%$ in NeaR-LLaMA and NeaR-GPT-4o. These results show that our adaptive thresholding performs better than a static threshold across a variety of MLLM choices, thus eliminating the need for tuning the hyperparameter $\tau$.

| Method | Bird-200 | | Car-196 | | Dog-120 | | Flower-102 | | Pet-37 | | Average | |
|---|---|---|---|---|---|---|---|---|---|---|---|---|
| | cACC | sACC | cACC | sACC | cACC | sACC | cACC | sACC | cACC | sACC | cACC | sACC |
| NeaR-LLaMA ($\tau$=0.5) | 51.7 | 70.5 | 53.3 | 60.5 | 59.0 | 68.6 | 78.1 | 61.7 | 82.2 | 85.9 | 64.8 | 69.5 |
| NeaR-LLaMA | 51.0 | 70.2 | 52.6 | 60.9 | 59.2 | 70.2 | 78.6 | 61.7 | 83.5 | 86.2 | **65.0 (+0.2%)** | **69.8 (+0.3%)** |
| NeaR-GPT-4o ($\tau$=0.5) | 54.7 | 74.5 | 57.9 | 59.7 | 62.1 | 74.6 | 79.6 | 52.1 | 83.0 | 83.8 | 67.4 | 68.9 |
| NeaR-GPT-4o | 55.8 | 75.6 | 57.0 | 60.0 | 61.6 | 74.4 | 80.6 | 52.1 | 82.9 | 84.0 | **67.6 (+0.2%)** | **69.2 (+0.3%)** |
| NeaR-GeminiPro ($\tau$=0.5) | 52.8 | 73.5 | 53.7 | 61.1 | 64.8 | 75.2 | 77.6 | 53.3 | 77.7 | 81.0 | 65.3 | 68.8 |
| NeaR-GeminiPro | 55.9 | 76.0 | 54.9 | 61.1 | 64.7 | 75.4 | 77.9 | 53.2 | 79.4 | 80.8 | **66.6 (+1.3%)** | **69.3 (+0.5%)** |
| NeaR-Qwen2 ($\tau$=0.5) | 34.3 | 65.0 | 57.0 | 64.0 | 55.2 | 71.3 | 74.0 | 61.4 | 74.1 | 76.4 | 58.9 | 67.6 |
| NeaR-Qwen2 | 35.5 | 65.6 | 58.0 | 64.0 | 56.6 | 71.7 | 75.8 | 62.5 | 73.8 | 76.4 | **60.0 (+1.1%)** | **68.0 (+0.4%)** |

Table A17: Evaluation of our dynamic threshold $\tau$ across different MLLMs compared to a static threshold $\tau = 0.5$. The use of a dynamic threshold shows consistent improvements across all MLLMs, with gains in NeaR-GeminiPro and NeaR-Qwen2, achieving increases of 1.6% and 1.1% in average cACC, respectively, and a minor gain of 0.2% in other cases. These results support the design choice to avoid the hyperparameter $\tau$, which can vary slightly across MLLMs.

### A8.7 Comparison with clustering based methods.

We compare against three clustering baselines discussed in FineR (Liu et al., 2024a): (i) K-Means (Ahmed et al., 2020) clustering on CLIP features, (ii) Sinkhorn-based parametric clustering (Caron et al., 2020) using CLIP and DINO features, and (iii) SCD (Han et al., 2023), which performs non-parametric clustering followed by CLIP-based narrowing of a large vocabulary consisting of 119k WordNet nouns and 11k bird names from Wikipedia. Results are reported in Table A18. We observe that NeaR outperforms classical clustering methods by a large margin using GPT-4o labels. Despite having the knowledge of the number of classes, the resulting cACC of K-Means algorithm is only 36.7%, showing that knowledge of class count alone does not yield high performance. The fact that alternative clustering variants built on different feature extractors achieve only marginal improvements further confirms that our performance gains come from the method design itself – not from any prior knowledge of the class count.

| Method | Avg. (cACC) |
|---|---|
| K-Means | 36.7 |
| CLIP-Sinkhorn | 21.6 |
| DINO-Sinkhorn | 19.1 |
| SCD | 52.2 |
| FineR | 57.0 |
| NeaR | **67.6** |

Table A18: Comparison of average clustering accuracy (cACC) across methods.

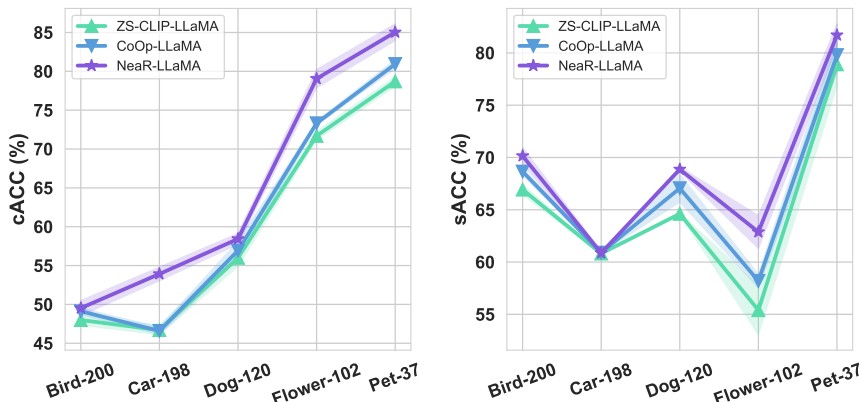

Figure A5: We report cACC and sACC under the effect of random sampling of training images across five datasets. The plot demonstrates minimal variance across datasets, highlighting the robustness of NeaR to variations in data selection.

### A8.8 Performance across variations in different few-shot splits

We have conducted extensive experiments across three random seeds to evaluate the robustness and consistency of our approach as shown in Figure A5. Specifically, for each seed, we sampled a unique set of $m$ images per class, ensuring diversity in the training data distribution across different runs. The results consistently demonstrate that NeaR outperforms the baseline approaches across various independent samplings, with minimum variance across datasets, indicating its stability and generalizability. This robustness across different random seeds highlights the effectiveness of our approach in handling variations in training data selection, further strengthening its practical applicability in real-world scenarios.

### A8.9 Comparison with other Pseudo-labeling based Methods

In this section we compare NeaR against two pseudo-labeling based methods DualCoOp(Sun et al., 2022) & VLPL(Xing et al., 2023). DualCoOp operates in a multi-label setting, where each image can belong to multiple classes, and the goal is to maximize activation across all true labels. On the other hand, in NeaR we operate in a problem setting where each image comes with a noisy label and we tend to find the best possible semantic label using the proposed pipeline. VLPL operates in a single-positive multi-label setting, where only one ground-truth label is provided per image, but the model is expected to predict multiple relevant labels. It leverages VLMs to generate pseudo-labels that enrich the supervision signal during training. In contrast, NeaR operates in a noisy single-label setting, where each image is associated with a possibility of semantically incorrect label from an MLLM. Rather than expanding to multiple labels, NeaR focuses on identifying the most semantically accurate label. We implemented DualCoOp and VLPL for our problem setting. The results are shown in Table A19 for LLaMA generated labels. As anticipated, given their fundamentally different objectives, both DualCoOp and VLPL underperform compared to NeaR.

| Method | Bird-200 | | Car-196 | | Dog-120 | | Flower-102 | | Pet-37 | | Average | |
|---|---|---|---|---|---|---|---|---|---|---|---|---|
| | cACC | sACC | cACC | sACC | cACC | sACC | cACC | sACC | cACC | sACC | cACC | sACC |
| DualCoOp | 35.6 | 31.7 | 20.8 | 45.2 | 41.9 | 45.5 | 6.3 | 27.8 | 18.5 | 48.6 | 24.6 | 39.7 |
| VLPL | 35.2 | 36.2 | 23.4 | 49.8 | 37.6 | 38.4 | 5.2 | 34.1 | 30.2 | 46.1 | 26.3 | 40.9 |
| NeaR-LLaMA | 51.0 | 70.2 | 52.6 | 60.9 | 59.2 | 70.2 | 78.6 | 61.7 | 83.4 | 86.1 | 65.0 | 69.8 |

Table A19: Comparison against pseudo-labeling based methods for labels generated by LLaMA3.2.

### A8.10 Analysis of retrieved images.

We perform qualitative analysis on the top three retrieved images ($\kappa = 3$) for the Flower-102 and Bird-200 datasets in Figure A6 and A7. In the results, each row shows a reference image on the left, followed by its

two nearest neighbors obtained using CLIP ViT-B/16 features. We show cases with both successful and unsuccessful neighbors. Successful neighbors are the ones which help in forming desired candidate set by including the ground-truth label. Unsuccessful candidate sets are the ones with no ground-truths.

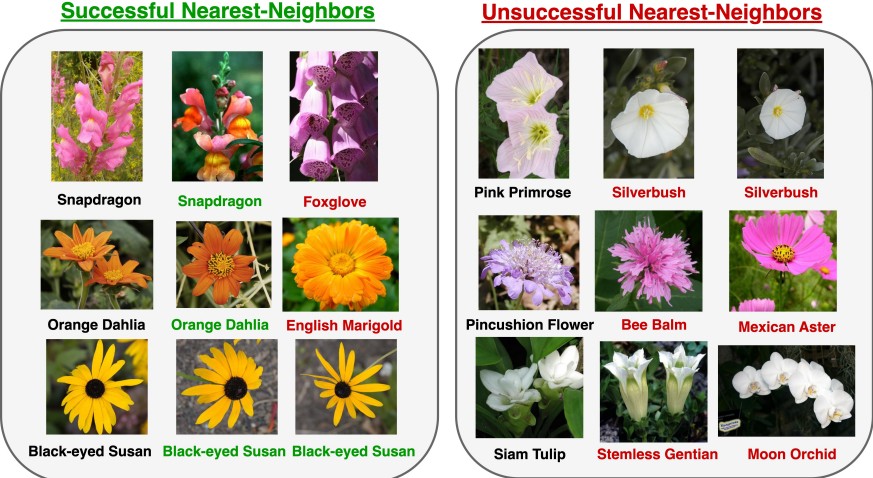

Figure A6: Qualitative results showcasing successful and unsuccessful neighbors in Flower-102 dataset.

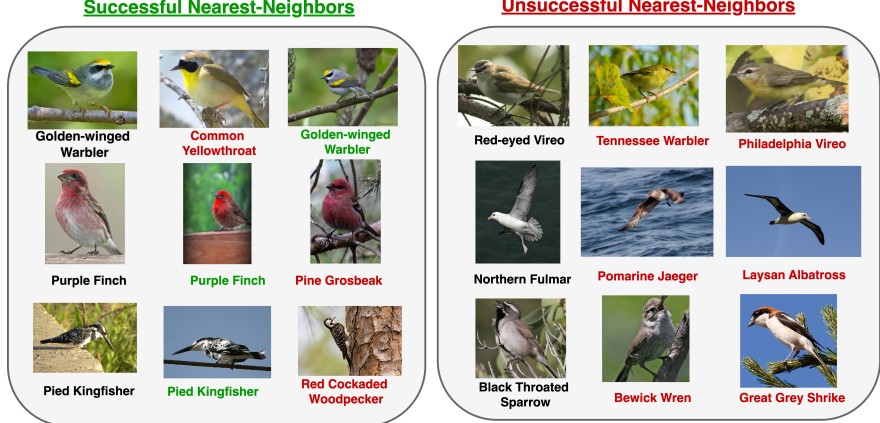

Figure A7: Qualitative results showcasing successful and unsuccessful neighbors in Bird-200 dataset.

### A8.11    Visualizing noisy samples from GMM

The qualitative results for visualizing noisy samples are shown in Figure A8. The figure displays several training examples the GMM marked as noisy immediately after warm-up. For each such image, we provide the ground-truth label, the model prediction (after the warm-up step), and the final post-training prediction. It can be clearly seen that the predictions post training have become much more semantically inclined and better than the initial predictions. We however cannot directly draw any comparison between the train set predictions and the test set predictions due to the label filtering step post training (the label space changes during inference). To make a fair comparison, we also study the scenario where we compare the test sample predictions of our model with the no-GMM ablation in Figure A9. We observe that the predictions of no-GMM ablation are more noisy.

**Flower-102 Dataset**

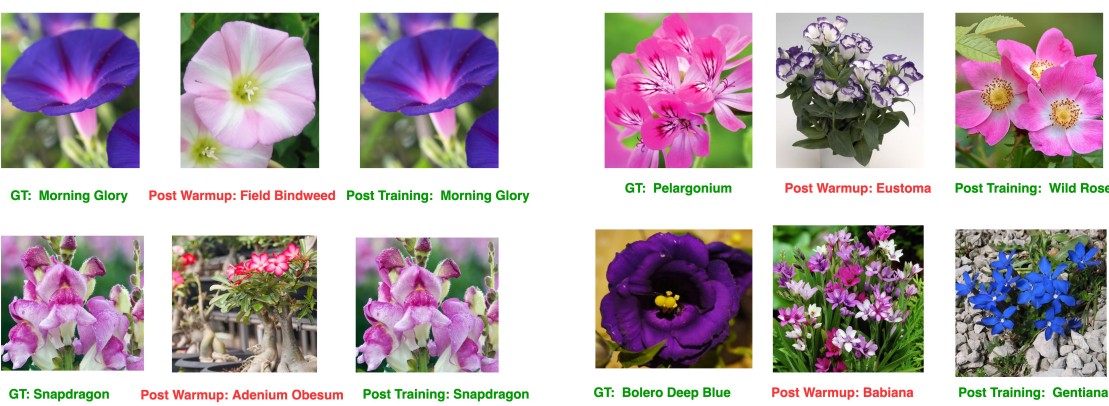

**GT: Morning Glory**   **Post Warmup: Field Bindweed**   **Post Training: Morning Glory**   **GT: Pelargonium**   **Post Warmup: Eustoma**   **Post Training: Wild Rose**

**GT: Snapdragon**   **Post Warmup: Adenium Obesum**   **Post Training: Snapdragon**   **GT: Bolero Deep Blue**   **Post Warmup: Babiana**   **Post Training: Gentiana**

Figure A8: Qualitative results of training samples classified as noisy by the GMM just after warmup and after full training.

**Flower-102 Dataset**

**GT: Barbeton Daisy**   **W/ GMM: Gerbara**   **W/o GMM: Tithonia**   **GT: Cyclamen**   **W/ GMM: Babiana**   **W/o GMM: Wood Anemone**

**GT: Colts foot**   **W/ GMM: Colts foot**   **W/o GMM: Taraxacum**   **GT: Canna Lily**   **W/ GMM: Canna Lily**   **W/o GMM: Iris Germanica**

Figure A9: Qualitative comparison of test image predictions between ours (with GMM) and no GMM ablation (w/o GMM).

### A8.12 Results on Other Open-Source MLLMs

In order to study the impact of NeaR on other open-source MLLMs, we query two weaker open-source MLLMs, LLaVA-1.5 (Liu et al., 2023)and BLIP2 (Li et al., 2023), to generate labels for our datasets. In the context of addressing the VF-FGVR problem, we observe that these models produce generic labels that lack fine-grained detail. For instance, in the Bird-200 dataset, images from various fine-grained classes such as American Goldfinch, Tropical Kingbird, Blue-headed Vireo, Yellow-throated Vireo, Blue-winged Warbler, Canada Warbler, Cape-May Warbler, and Palm Warbler were all labeled simply as 'Bird' by LLaVA. This lack of specificity results in a low cACC of 9.8% for CoOp-LLaVA and 4.7% for NeaR-LLaVA. This trend is also observed with BLIP2. The inability of these MLLMs to generate diverse fine-grained labels makes them a poor choice to solve the VF-FGVR task.

### A8.13 Prompts used to generate labels from MLLMs.

In Table A20, we describe the prompts used to obtain the labels for both proprietary and open-source MLLMs. We give different prompts for different datasets. As part of future work, we would like to explore if different prompting strategies can give better labels.

| MLLM | Prompt Structure |
|---|---|
| GPT-4o, GeminiPro | "You are a multimodal AI trained to provide the best fine-grained class label for a given \<dataset> image. Provide the best fine-grained class label for the given \<dataset> image.  Do not return anything else.", \ |
| LLaMA, Qwen | "Give me a fine-grained label for this \<dataset>. For example, \<samplelabel>. Just print the label and nothing else.", \ |

Table A20: A summary of prompts used for querying MLLM models used in this paper. In these prompts, $\texttt{dataset} \in \{bird, car, dog, flower, pet\}$. `img` refers to the image being queried for fine-grained class label. `Samplelabel` for *bird* is `Black Throated Sunbird`, `samplelabel` for *car* is `2012 BMW M3 coupe`, etc. We observe that open-source models like LLaMA require extra supervision in terms of sample labels for better performance.

