# OpenReview forum: "Efficient Vocabulary-Free Fine-Grained Visual Recognition in the Age of Multimodal LLMs"
_TMLR — Accepted by TMLR_

### Review · Reviewer_Q8Kb · 2025-06-11

**Summary Of Contributions:**

This paper proposes a framework for vocabulary-free, fine-grained visual recognition. Pseudo-labels, generated from Multimodal Large Language Models (MLLMs), are refined and enhanced by their k-Nearest Neighbors (k-NN) sets during training. These labels are then used to supervise a CLIP model. Additionally, techniques such as a warmup phase and candidate set-guided label refinement are employed to improve overall performance. A label filtering method is also proposed for further enhancement at test time. The proposed method demonstrates greater efficiency compared to MLLM-based baselines.

**Audience:**

Yes

**Claims And Evidence:**

Yes

**Requested Changes:**

Please address the concerns raised in the weaknesses section.

**Strengths And Weaknesses:**

**Strengths**

1. The proposed method shows improved efficiency compared to MLLM-based baselines and better performance than contemporary VF baselines.

2. The proposed method is computationally efficient and straightforward to implement.

**Weaknesses**

1. **The presentation could be improved:**\
    (1).  Several symbols and concepts are not clearly explained. These include: the random candidate set method mentioned in Table 2, the implementation details of prompt training in the warmup subsection, the construction of $q^j_i$ (specifically, how a set is converted into a score), and the circle-dot symbol used in the probability rescaling procedure.

2. **The ablation studies are not comprehensive enough:** \
 (1) Ablations on the choice of vision foundation model (e.g., DINO, MAE) for selecting the top-k similar images are not included.  \
 (2) An ablation study for the value of k in the top-k selection is missing. \
 (3) The paper is missing an ablation study for the proposed candidate set-guided label refinement strategy. \
 (4) The primary ablation results are presented on a single dataset. It is recommended to perform these ablations on all evaluated datasets.

3. **The claim of improved efficiency is only supported by data in Table 1**; efficiency metrics for the results in Table 3 are missing.

4. **Some claims appear to be inappropriate or incorrect.** For example, the authors state in the contributions that their method outperforms MLLM baselines. However, the results in Table 3 show that the best overall performance is achieved by the Qwen2-VL-7B model, which is an MLLM.

5. The scaling capabilities of the method on larger CLIP backbones (e.g., Large and Giant) are not validated. Furthermore, the generalization of the method to CLIP variants (e.g., SigLIP) has not been explored.

---

> ### Author Response · Authors · 2025-07-10
> **Response to Reviewer Q8Kb**
>
> > Presentation could be improved
>
> We thank the reviewer for pointing these out. In Table 2, to generate the random candidate set we pick two random labels (for $\kappa=3$) from the set of all MLLM generated labels. As described in Implementation Details para of Sec 4, we follow CoOp to add 16 learnable text prompts. The same set of prompts are optimized during the warmup stage, and for the subsequent training phase. We describe construction of q in Sec 3.3 under "Candidate Set Guided Label Refinement". The $\odot$ symbol represents the hadamard product. We have added these clarifications to the revised version and believe they improve readability.
>
> > Ablation Studies are not comprehensive enough
>
> 1. DINO and MAE backbones
>
> As suggested, we use DINO and MAE models solely for the nearest neighbor candidate set construction step, while retaining our default ViT-B/16 CLIP backbone for prompt training. The results for LLaMA-3.2, presented below, show that our method continues to achieve competitive performance even when the k-NN step is based on features from other backbones. We have added this discussion to Sec A8.6 of the revised manuscript.
>
> | Method| Bird || Car|| Dog || Flower | | Pet ||Avg|Avg|
> |-|-|-|-|-|-|-|-|-|-|-|-|-|
> || cACC|sACC|cACC|sACC|cACC|sACC|cACC|sACC|cACC|sACC|cACC|sACC|
> |DINO|50.3|70.5|50.6|61.8|60.4|72.6|79.7|60.8|82.4|88.3|64.7|70.8|
> |MAE|48.6|70.1|50.9|61.5|59.9|72.2|76.6|59.8|84.8|88.6|64.2|70.4|
> |CLIP ViT-B/16|51.03|70.2|52.6|60.9|59.2|70.2|78.6|61.7|83.4|86.1|65.0|69.8|
>
> 2. k ablation
>
> We would like to kindly bring to your attention that the ablation on $\kappa$ is already provided in Appendix Sec A8.11. In the revised version we extend our analysis by varying $\kappa$ when the number of training shots $m$ is randomly varied between $1$ to $10$ for each class. The results are shown in Table 4 of response to reviewer 6edX and shows that performance is not very sensitive to $\kappa$, and justifies our practical choice of $\kappa=3$.We add this discussion to Sec A8.11 of the Appendix in the revised version.
>
> 3. and 4. Ablation of label refinement across all datasets; ablation on all datasets
>
> Thank you for bringing this up. We conduct an ablation of the various components of the label-refinement step. The table below shows the results of our ablation for all datasets. The first row shows the result of not performing label-mixup, which shows a drop of $2\%$ avg cACC. As Label Mixup blends the model’s predicted label with information from the one-hot label or candidate set, and removal of it results in the model solely relying on the MLLM's raw prediction and loses the helpful prior information. In the second row we do not rescale the refined labels of noisy samples, leading to a much higher drop in cACC. As Rescaling ensures that the refined label has non-zero probabilities only for the candidate set, enabling it to learn from a relevant set of candidate labels and removal of this leads to a misleading label signal for the noisy samples. In the third row we study the effect of not using the candidate set for noisy samples (i.e using $y_i$ instead of $q_i$ for noisy samples), which leads to a drop of $2.5%$ in avg cACC. The best performance is obtained when all components of our proposed label-refinement are present. This discussion has been added to Sec 4.2 in the revised manuscript.
>
> |Label Mixup|Rescaling|Candidate Set|Bird || Car|| Dog|| Flower | | Pet || Avg|Avg|
> |-|-|-|-|-|-|-|-|-|-|-|-|-|-|-|
> ||||cACC|sACC|cACC|sACC|cACC|sACC|cACC|sACC|cACC|sACC|cACC|sACC|
> |No|Yes|Yes|51.3|71.3|48.6|60.7|59.4|71.4|75.9|60.1|79.7|84.0|63.0|69.5|
> |Yes|No|Yes|49.1|69.4|50.9|60.6|59.5|71.8|77.1|61.4|77.8|84.1|62.9|69.5|
> |Yes|Yes|No|48.5|69.1|48.9|59.7|57.1|69.6|76.9|60.1|81.8|84.1|62.6|68.5|
> |Yes|Yes|Yes|51.03|70.2|52.6|60.9|59.2|70.2|78.6|61.7|83.4|86.1|65.0|69.8|
>
>
> For completion sake we perform an ablation study of all components in our framework (of which only a subset is shown here). The results are shown in Tab 2 of reviewer 6edX, and corroborate our design choices.
>
> > Claims of Improved efficiency missing in Table 3
>
> Table 1 comprehensively covers the different MLLM types and all baselines we designed. For instance, we report inference time, cost, and cACC for GPT-4o, which closely reflects performance for Gemini Pro as well. Similarly, results for LLaMA are representative of Qwen. For ZS-CLIP-MLLM, CoOp-MLLM, and NeaR-MLLM, inference time and cost remain the same, with only a slight overhead in training time. Thus the metrics in Table 1 also hold for the other MLLMs studied in Table 3. We have added a discussion on the efficieny of NeaR in Appendix A8.1 of the revision.

---

> > ### Author Response · Authors · 2025-07-10
> > **(2) Cont. Response to Reviewer Q8Kb**
> >
> > > Some claims are inappropriate
> >
> > The impression that MLLMs perform better may arise from the sACC metric. However, we would like to emphasize that cACC is a more informative and stricter metric, as it captures the consistency of assigning semantically coherent labels across similar instances. While MLLM-generated labels often appear relevant, they may not be semantically apt sometimes. Thus, a high sACC indicates general label quality, but the real challenge lies in achieving high cACC. When comparing cACC, our method achieves significant gains of 8.4%, 9.8%, and 16.6% over GPT-4o, Gemini Pro, and LLaMA3.2, and is competitive with Qwen2-VL with a lower inference time.
> >
> > > Different CLIP backbones
> >
> > We present results using ViT-L and ViT-G backbones below. We observe performance gains of 5.5% in cACC and 3.2% in sACC with ViT-L, and gains of 6.9% in cACC and 2.9% in sACC with ViT-G. Despite these improvements, our method still outperforms CoOp-LLaMA by 3.2% with ViT-L and 4.6% with ViT-G, demonstrating its effectiveness across stronger backbone architectures. We also experiment with the SigLIP ViT-B/16 backbone which has a different pretraining objective compared to CLIP. Our results indicate that our method does not transfer effectively to SigLIP. We conjecture that our method fails because our final objective resembles the InfoNCE objective used by CLIP, and is different from the Sigmoidal loss employed by SigLIP. However, we are competitive with the CoOp baseline in avg cACC. We have added this discussion in Sec A8.7 in revised manuscript.
> >
> > | Method| Bird || Car || Dog || Flower | | Pet ||Avg|Avg|
> > |-|-|-|-|-|-|-|-|-|-|-|-|-|
> > || cACC|sACC|cACC|sACC|cACC|sACC|cACC|sACC|cACC|sACC|cACC|sACC|
> > |CoOp-LLaMA (ViT-L)|55.8|73.4|51.1|63.0|65.1|74.3|81.7|63.3|83.2|87.6|67.3|72.3|
> > |NeaR-LLaMA (ViT-L)|54.5|73.1|62.5|62.3|67.9|74.6|82.4|64.7|84.9|87.3|70.5|72.4|
> > |CoOp-LLaMA (ViT-G)|56.2|74.0|51.2|63.1|64.7|73.8|82.2|63.1|82.4|87.5|67.3|72.3|
> > |NeaR-LLaMA (ViT-G)|59.0|74.9|63.0|61.7|66.2|74.2|87.5|64.9|83.7|85.0|71.9|72.1|
> > |CoOp-LLaMA (SigLIP)|36.8|64.1|61.9|62.8|58.5|70.2|71.1|57.2|75.6|80.0|60.8|66.9|
> > |NeaR-LLaMA (SigLIP)|34.0|63.0|71.5|62.9|57.7|70.2|70.9|56.7|72.9|78.6|61.4|66.3|

---

### Review · Reviewer_xzFU · 2025-06-15

**Summary Of Contributions:**

1. The proposed method enables CLIP-like models to perform vocabulary-free zero-shot image classification using labels generated from a Multimodal LLM (MLLM).
2. The proposed method mitigates noisy labels generated by the MLLM by aggregating labels of similar images retrieved using CLIP's image features.
3. Extensive experiments are done including various open-source and proprietary MLLMs, along with ablation studies showcasing the superior performance of the proposed model.

**Audience:**

Yes

**Claims And Evidence:**

Yes

**Requested Changes:**

1. I would like to see an ablation that justifies the order of operations: 1) Nearest Neighbor selection; 2) GMM; 3) Label Filtering. Also, in Table-5, the paper should include results with and without GMM.
2. The paper should include an ablation study showing performance of NeaR with and without prompt tuning.
3. The authors should also include results of prompt tuning with ZS-CLIP-GT (Upper Bound) method. I believe that would denote the true upper-bound performance of CLIP.
4. The paper should cite and discuss other pseudo-label based classification methods such as DualCoOp [1], VLPL [2] and CaSED [3]. Ideally, I would like to see NeaR compared with these methods especially VLPL, which is very similar to proposed method.
5. The paper needs to visualize the noisy samples filtered out by the GMM method. Do these sample correlate with the failure cases of the model?

References

[1] Sun, Ximeng, Ping Hu, and Kate Saenko. "Dualcoop: Fast adaptation to multi-label recognition with limited annotations." Advances in Neural Information Processing Systems 35 (2022): 30569-30582.

[2] Xing, Xin, et al. "Vision-Language Pseudo-Labels for Single-Positive Multi-Label Learning." 2024 IEEE/CVF Conference on Computer Vision and Pattern Recognition Workshops (CVPRW). IEEE, 2024.

[3] Conti, Alessandro, et al. "Vocabulary-free image classification." Advances in Neural Information Processing Systems 36 (2023): 30662-30680.

**Strengths And Weaknesses:**

Strengths

1. The proposed method achieves vocabulary-free classification with performance comparable to known label setting for CLIP-like models, although without any groundbreaking results or discoveries.

Weaknesses

1. The paper borrows and stitches ideas from various existing works. The only novelty being the use of MLLM for generating labels.
2. It is difficult to understand the correct flow of the entire methodology. There are three distinct operations: 1) building the candidate set with MLLM and nearest neighbor image retrieval; 2) GMM for filtering out noisy samples; 3) Label filtering. My question is, does the order in which the operations are performed matters? Is GMM performed before label filtering? I guess the GMM and label filter operations are highly correlated.
3. I don't agree with the validity of experiment done in Figure-A4. The authors select 9-shot data from the Flowers-102 dataset, and ablate on the value of k from 1 to 9. However, in the real world, one might not be guaranteed to have 9 examples for each class. One experiment done in Figure-2 uses fixed k for varying shots. A complement to these results would be a single graph that shows the performance of models with simultaneously varying k and m values.
4. The paper misses discussion on the similarities and dissimilarities of their method with retrieval-augmented classification methods such as RAC [1].
5. Please see `Requested Changes` section for further comments.

References

[1] Long, Alexander, et al. "Retrieval augmented classification for long-tail visual recognition." Proceedings of the IEEE/CVF conference on computer vision and pattern recognition. 2022.

---

> ### Author Response · Authors · 2025-07-10
> **(1) Response to Reviewer xzFU**
>
> We thank the reviewer for their thoughtful comments. We respond to each comment below.
>
> > Paper borrows ideas from existing works
>
> We respectfully disagree with the characterization of our work as a simple combination of existing ideas. While we leverage established components such as candidate set construction, prompt tuning, and learning with noisy labels (LNL), our key contribution lies in the careful integration of these elements into a unified framework that directly addresses the significant challenge of interpreting noisy, uncurated labels generated by MLLMs for fine-grained recognition. The components are carefully designed to tackle the semantic noise amongst the labels which other baselines failed to handle. We agree our method is simple in its working, but is effective.
>
> > Unclear about the pipeline
>
> We believe there may have been some confusion regarding the methodological flow. We agree that GMM-based filtering and label refinement are closely related, and we clarify their roles below:
>
> In our method, the three components — candidate set construction, Label refinement, and Label Filtering—serve distinct and sequential roles. We begin by constructing the candidate sets by computing the nearest neighbors for training samples using the frozen CLIP vision encoder. We perform a one-time warmup step that aides in disambiguation of images into clean and noisily labeled set as described in Sec 3.3. Note that candidate set construction and warm-up are one-time steps. Post warmup, for each epoch, we compute the training loss for each sample and apply a Gaussian Mixture Model (GMM) to separate training images into clean and noisily labeled subsets. Clean samples are trained using standard cross-entropy on the MLLM label, while noisy samples are trained using candidate set supervision. After complete training, we apply Label Filtering as a global post-processing step: we retain only those labels that appear both as the model’s final predicted argmax over all labels and as the argmax within the sample’s candidate set. This ensures that the final prediction vocabulary consists of labels that are both supported by the training process and confidently predicted.
> For the ablation on the order of operations, our method follows the sequence: Candidate set construction (CS) -> GMM -> Label Filtering (LF). We believe other permutations are less meaningful given the roles each component plays. One alternative could be GMM -> CS -> LF, since the GMM split is computed independently of the candidate sets (i.e it uses model loss computed on just the MLLM labels, without involving candidate sets in the loss). As the outcome of the CS step is same every time, it unnecessarily results in computational overhead.
>
> We thank the reviewer for raising this insightful question. To generalize this idea, we conduct an experiment where m is randomly sampled between 1 and 10 ([Random sampling of Flower-102 dataset](https://anonymous.4open.science/r/NeaR-DC02/flowers.png)), and we ablate over different values of k ranging from 1 to 9. We find that the cACC remains relatively stable across this range, with only minor variations. This indicates that our method is not particularly sensitive to the choice of k, which can be selected flexibly. In our implementation, we use k=3 as a practical default. The results for this ablation on the Flowers-102 dataset are shown in the table below.  We have added this in Sec A8.11 in the revised manuscript.
>
> | Method| Flower-102 ||
> |-|-|-|
> ||cACC|sACC|
> |k=1|69.8|50.1|
> |k=2|72.3|51.7|
> |k=3|73.5|52.0|
> |k=4|74.7|53.9|
> |k=5|74.0|53.1|
> |k=6|74.3|53.6|
> |k=7|73.6|53.8|
> |k=8|71.5|52.6|
> |k=9|70.9|52.0|
>
> > Comparison with retrieval-augmented classification method-RAC
>
> In Tab 3 of the paper we already compare our method against a Retrieval-based method called RAR[1] which uses a multimodal retriever with external memory, retrieving and ranking top-k samples using an LLM. We outperform RAR by $6.5\%$ in cACC and $4.5\%$ in sACC. As requested, we also compare our method with RAC(Long et al., CVPR 2022). Both NeaR and RAC leverage retrieval, their objectives and settings are fundamentally different. RAC uses retrieval to refine features via a learned transformer using a labeled memory bank, operating under full supervision for long-tail classification. In contrast, NeaR operates in a much more challenging weakly supervised setting, using noisy labels from MLLMs and retrieval purely to construct partial candidate label sets for training. Unlike RAC, NeaR does not assume access to ground-truth labels or a known class set, making it more suitable for open-vocabulary, fine-grained recognition. Furthermore, NeaR is lightweight, backbone-agnostic, and avoids complex memory-based architectures. We have added this discussion to Section 2 of the revised manuscript.
>
> [1] Ziyu Liu et al, Rar: Retrieving and ranking augmented mllms for visual recognition. arXiv preprint arXiv:2403.13805, 2024c.

---

> > ### Author Response · Authors · 2025-07-10
> > **(2) Cont. Response to Reviewer xzFU**
> >
> > > NeaR with and without Prompt tuning
> >
> > To answer this question we study the performance on NeaR for an alternative fine-tuning method. We follow CLIP-Adapter[2] to finetune linear adapters added on top of both the frozen CLIP image and text encoders, as an alternative to prompt-tuning.
> > We begin by noting that prompt learning is more efficient in terms of param count. In our setup, we use 16 prompt tokens for a total of 16x512 learnable params, whereas a fully-connected linear layer has 512*K params where K is the number of labels/classes (eg. number of GPT-4o generated labels is K=300 for Bird-200). The table below shows cACC results for $\alpha=0.8$ (from [2]) for GPT-4o generated labels. We observe that NeaR_Adapter-GPT4o outperforms the adapter baseline (59.98% vs 48.75% average cACC), but still falls short of the performance achieved by our prompt-tuning based NeaR-GPT4o (59.98% vs 67.6% average cACC). This suggests that adapter based tuning is unsuitable for the VF-FGVR setting, potentially due to its reduced robustness to open-vocabulary label noise. This is consistent with findings from Wu et al. [3] which shows that prompt tuning is more robust to label noise compared to linear classifiers. We have added this discussion to Sec A8.9 in the revised manuscript.
> >
> > | Method| Bird|| Car || Dog|| Flower | | Pet ||Avg|Avg|
> > |-|-|-|-|-|-|-|-|-|-|-|-|-|
> > || cACC|sACC|cACC|sACC|cACC|sACC|cACC|sACC|cACC|sACC|cACC|sACC|
> > |NeaR w/o prompt tuning |50.2|73.9|48.0|59.2|54.7|69.7|70.2|49.3|76.6|82.3|59.9|66.9|
> > |NeaR w/ Prompt|51.03|70.2|52.6|60.9|59.2|70.2|78.6|61.7|83.4|86.1|65.0|69.8|
> >
> > [2] Gao, Peng et al. “CLIP-Adapter: Better Vision-Language Models with Feature Adapters.” ArXiv 2021.
> > [3] Wu, Cheng-En et al. “Why Is Prompt Tuning for Vision-Language Models Robust to Noisy Labels?” ICCV 2023.
> >
> > > Upper bound by training CoOp using GT
> >
> > We thank the reviewer for suggesting a meaningful upper bound. Using the same ViT-B/16 backbone configuration, we trained a CoOp-style model for 50 epochs using ground-truth labels. This upperbound has been added to Table 3 in the revised manuscript.
> >
> > > Additional baselines
> >
> > We thank the reviewer for suggesting other pseudo-labeled classification works--DualCOOp, VLPL and CaSED. DualCoOp operates in a multi-label setting, where each image can belong to multiple classes. On the other hand, in NeaR we operate in a problem setting where each image comes with a noisy label and we tend to find the best possible semantic label using the proposed pipeline. VLPL operates in a single-positive multi-label setting, where only one ground-truth label is provided per image, but the model is expected to predict multiple relevant labels. It leverages VLMs to generate pseudo-labels that enrich the supervision signal during training. Rather than expanding to multiple labels, NeaR focuses on identifying the most semantically accurate label. We already cite and compare with CaSED in our main table. As requested, for completeness sake, we ran the DualCoOp and VLPL for our problem setting. The results are as shown below for LLaMA labels. As anticipated, given their fundamentally different objectives, both DualCoOp and VLPL underperform compared to NeaR. We have added this discussion to Sec A8.10 in the revised manuscript.
> >
> > | Method| Bird || Car|| Dog|| Flower | | Pet ||Avg|Avg|
> > |-|-|-|-|-|-|-|-|-|-|-|-|-|
> > || cACC|sACC|cACC|sACC|cACC|sACC|cACC|sACC|cACC|sACC|cACC|sACC|
> > |DualCoOP|35.6|31.7|20.8|45.2|41.9|45.5|6.3|27.8|18.5|48.6|24.6|39.7|
> > |VLPL|35.2|36.2|23.4|49.87|37.6|38.4|5.2|34.1|30.2|46.1|26.3|40.9|
> > |CaSED|25.6|50.1|26.9|41.4|38.0|55.9|67.2|52.3|60.9|63.6|43.7|52.6|
> > NeaR-LLaMA|51.03|70.2|52.6|60.9|59.2|70.2|78.6|61.7|83.4|86.1|65.0|69.8|
> >
> > > Visualizing noisy samples from GMM and its correlation to failure cases
> >
> > We thank the reviewer for this interesting question. We show the qualitative results for Flower-102 dataset [here](https://anonymous.4open.science/r/NeaR-DC02/gmm_failure.png). The figure displays several training examples the GMM marked as noisy immediately after warm-up. For each such image, we provide the ground-truth label, the model prediction (just after warm-up step), and the final post-training prediction. It can be clearly seen that the prediction post training have become much more semantically inclined and better than the predictions just after warm-up. We cannot directly draw any comparison between the train set predictions and the test set predictions as both of them operate in a slightly different label space. The label space changes during inference (becomes less noisy) due to the label filtering step post training. To make a fair comparison, we also study the scenario where we compare the test sample predictions of our model vs the no-GMM ablation [here](https://anonymous.4open.science/r/NeaR-DC02/w_and_wo_gmm.jpg). We observe that the predictions of no-GMM ablation are more noisy. We have added this discussion to Sec A8.16 in the revised manuscript.

---

### Review · Reviewer_6edX · 2025-06-20

**Summary Of Contributions:**

The authors tackle the problem of fine-grained visual recognition in the vocabulary-free setting, i.e., where the target label list is unknown to the models. To solve the problem, they consider having access to a small balanced set of unlabelled images of the dataset and using a multimodal large language model (MLLM) to pseudo-label them. They introduce Nearest-Neighbor Label Refinement (NeaR) to fine-tune a model for the task in the presence of noisy labels, i.e., those generated by the MLLM, considering refinement and filtering steps to mitigate the noise in the pseudo-labels. Finally, they fine-tune the model with prompt tuning, improving the performance on five fine-grained datasets.

**Audience:**

Yes

**Broader Impact Concerns:**

No concerns

**Claims And Evidence:**

Yes

**Requested Changes:**

The authors should focus on addressing the weaknesses I reported above, with most of the focus on the following points:

- Discuss the possibility of their model "gaming" the cluster accuracy.
- Demonstrate that their approach works without the reliance on the k-shots balanced on the dataset classes, by showing the performance with, e.g., random sampling from the training set.
- Explain the mismatch between Tab. 2 and Tab. 3 and the ambiguities in the first.
- Report an ablation study on the construction of refined labels, to demonstrate how it contributes to the overall performance, and confirm whether the selected transformations are not overfitted to the data.
- Propose a hypothesis on why having more shots increases semantic accuracy and decreases cluster accuracy for most methods.

**Strengths And Weaknesses:**

**Strengths**

- The work is well-motivated and tackles a novel problem that is more practical than traditional zero-shot fine-grained visual recognition.
- The presentation is curated, and the content is clearly explained.
- The proposed approach consistently improves over the previous state-of-the-art.

**Weaknesses**

- The assumption of having exactly k samples per class (even without labels) is quite strong, as it implicitly requires knowledge of the class distribution during data selection, effectively using label information before applying the proposed method. Although the authors attempt to address this by considering a more "realistic" imbalanced setting, this scenario is still artificial: it assumes access to at least k samples from every class. This allows the total number of classes to be inferred simply by dividing the dataset size by k, which undermines the realism of the setup. Knowing or approximating the number of classes in advance enables the model to produce a similar number of output clusters, which can artificially boost cluster accuracy by ensuring a one-to-one mapping with ground-truth labels rather than a more challenging many-to-one or one-to-many situation. This may also be confirmed in Tab. 3, as the main gains are on cluster accuracy.
- Instead of the imbalanced setting, I believe it would be more interesting to show situations where the images are randomly sampled from the training set, without any assumption on the presence of all the classes in the selection process.
- There is a mismatch in the semantic similarity reported in Tab. 2 (i.e., the one demonstrating that k-nn filtering improves candidate selection) and the main results in Tab. 3. I understand that the difference is due to all the other components introduced to fine-tune the model after the initial selection procedure. However, this may highlight that the training is not optimized, as the semantic similarity shows an average reduction of approximately 8 points. Moreover, the average semantic similarity reported in Tab. 2 does not match any of the averages for the MLLMs in Tab. 3.
- The approach used to construct refined labels feels very overfitted to the data, as it uses a lot of transformations (also different between the clean and the noisy subsets). Moreover, I believe there are no ablations on this step of the pipeline.
- When increasing the number of shots, cluster accuracy decreases and semantic accuracy increases. Could the authors propose a hypothesis for this behaviour?
- The selected backbones in one ablation study (RN50, RN101, ViT-B/16, and ViT-B/32) form two closely related pairs, limiting the diversity of the analysis. It may be more informative to explore how performance varies when scaling within a single architecture family (e.g., ViT-B to ViT-L or ViT-g) or when using models with different pretraining strategies (e.g., SigLIP).
- Unclear how using the commercial models in Tab. 11 resulted in an out-of-memory issue. If the problem is the generation of too many candidates, then the authors could convert the problem into an image-to-text retrieval problem (similar to what CaSED did) and retrieve for each image only the top-k most similar candidates.


**Minor weaknesses**

- The authors state they are the first to explore using MLLMs for the task, but that may be incorrect, as FineR uses BLIP for generation, and CaSED has baselines with BLIP and ablations where they test generating the candidate list instead of doing retrieval.
- The authors should probably add spaces for some of the paragraphs (e.g., in the Related section, where there are some bold words to introduce a paragraph but no newline).
- I suspect the estimate of the costs for commercial models is not optimized, as the authors report ~100$ for 30k images. However, for these situations it is best to perform batched inference via the API, which costs ~5 times less and is guaranteed to complete in at most 24 hours.
- Tab. 2 does not explain which MLLM was used as a baseline.
- The default value of nearest-neighbours and the number of shots is reported twice in the implementation details. Also, the decision for k=3 is very arbitrary, as it is more common to have 1, 2, 4, 8, 16 shots in few-shot settings.
- The reported relative improvements are difficult to interpret, as the tables use inconsistent reference baselines. In some cases, the reference is not the second-best performing method, which can exaggerate the perceived effectiveness of the proposed approach.
- Tab. 5 has two columns (warmup, GMM) which are never ablated. I advise the authors to either remove them or (better) report the ablation also without those components.
- About the imbalanced training setting, I am unsure if the scenario can be called "long-tailed" due to the range of occurrences of classes, i..e, 3-10.
- Unclear why the qualitative are against an MLLM and not against baselines (e.g., FineR).

---

> ### Author Response · Authors · 2025-07-10
> **(1) Response to Reviewer 6edX**
>
> We thank the reviewer for their thoughtful and comprehensive comments. We address each comment below.
>
> > **Weaknesses**: The assumption of having exactly k samples per class (even without labels)... gains on cluster accuracy. Instead of the imbalanced setting, I believe ... the selection process.
> > **requested changes** gaming the cluster accuracy
> > **requested changes** Demonstrate that their approach works ... random sampling from training set
>
> We respectfully but strongly disagree with the claim that our method leverages knowledge of the number of classes—either directly or indirectly—to improve cACC. Our experimental protocol strictly follows that of FineR, using three images per class as the default setting and adopting the same evaluation metrics (cACC and sACC). Consistent with FineR, our method has no access to the number of classes during training. Furthermore, we consistently outperform the CoOp baseline which also operates in the same setup. Additionally, in Table A18, we provide results from a pure clustering baseline where the number of classes is known in advance where we apply a simple k-means clustering on CLIP features. Despite having the knowledge, the resulting cACC is only 36.7%, showing that knowledge of class count alone does not yield high performance. Thank you for your suggestion of studying performance under random sampling of the training set. As requested, we present results under a random data distribution, where the number of samples per class ranges from 1 to 10. The data distribution for each dataset is shown [here](https://anonymous.4open.science/r/NeaR-DC02/random_sampling.png). Our method outperforms CoOp-LLaMA by 8.3% in cACC and 1.4% in sACC, demonstrating its robustness to data imbalance. We add this discussion to the Appendix Sec A8.2 in the revised version.
>
> | Method| Bird || Car|| Dog || Flower | | Pet ||Avg|Avg|
> |-|-|-|-|-|-|-|-|-|-|-|-|-|
> || cACC|sACC|cACC|sACC|cACC|sACC|cACC|sACC|cACC|sACC|cACC|sACC|
> |CoOP-LLaMA|42.1|52.8|42.0|56.7|42.1|54.0|60.3|43.6|54.1|56.3|48.1|52.7|
> |Ours-LLaMA|43.0|52.0|47.8|55.3|55.8|55.7|73.5|52.0|61.9|55.7|56.4|54.1|
>
>
> To further validate the robustness and versatility of our approach, we study the performance of NeaR under various setups: (i) Performance under long-tail training data (Table 6), random data distribution (Table A9); (ii) Using different backbones such as CLIP RN, CLIP ViT-G etc. (Table 7 \& Table A14); (iii) Different prompting strategies (Table A15); (iv) Varying number of nearest neighbors $\kappa$ (Table A18, Figure A5); (v) Under different few-shot splits (Figure A7).
>
> > **weaknesses**: The approach used to construct refined labels ... step of the pipeline
> > **requested changes**: Report an ablation study on the construction of refined labels ... overfitted to the data.
>
> As detailed in Sec 3.3, our intuition is that samples classified as "noisily labeled" cannot rely on their MLLM generated label. We instead rely on the candidate set $q$ (rather than the MLLM label $y$) for supervision.
>
> We conducted a thorough ablation to evaluate the contribution of each component in our pipeline. We split the components of "Candidate Set Guided Label Refinement" into label mixup, rescaling, and using the candidate set guidance.
> The best performance is observed when all components — including warmup, GMM-based partitioning, label mixup, rescaling, candidate set guidance, and post-training label filtering, are used. This configuration yields the highest average cACC (65.0%) and sACC (69.8%), re-confirming the effectiveness of proposed approach.
>
> Removing the warm-up phase leads to a noticeable drop in performance, as it causes the GMM to produce a less reliable clean/noisy split. GMM is important because it splits the data into clean and noisily labeled samples. Without this split, the candidate set guidance is no longer used and only the MLLM generated labels are used throughout. Label mixup blends the label (one-hot or candidate set) with the model’s prediction, and acts as a regularizer. Rescaling ensures that the refined label for a noisy sample has non-zero probabilities only for the candidate set, enabling it to learn from a relevant set of candidate labels. Removal of this leads to a misleading label signal for the noisy samples. Disabling either candidate set guidance or label filtering leads to noticeable drops across most datasets. To study the effect of removal of the candidate set guidance, we refine the labels of the noisy samples as
> $\text{shrp}(f_{\theta}(x_{i}), T)$, i.e we only used the sharpened CLIP pseudolabel. We also remove the candidate set based filtering $F_{cand}$, as defined in Sec4. These ablations confirm that our components which are simple but carefully designed are integral for the performance gains. We have added this to the Sec4.2 in the revised manuscript.

---

> > ### Author Response · Authors · 2025-07-10
> > **(2) Cont. Response to Reviewer 6edX**
> >
> > |Warmup|GMM|Label Mixup|Rescaling|candidate Set|Label Filtering| Bird || Car || Dog || Flower | | Pet || Avg|Avg|
> > |-|-|-|-|-|-|-|-|-|-|-|-|-|-|-|-|-|-|
> > |
> > |||||||cACC|sACC|cACC|sACC|cACC|sACC|cACC|sACC|cACC|sACC|cACC|sACC|
> > |No|Yes|Yes|Yes|Yes|Yes|48.81|71.5|49.8|61.9|56.8|71.9|75.4|61.3|79.3|85.4|62.0|70.4|
> > |Yes|No|Yes|Yes|No|Yes|48.5|69.1|48.9|59.7|57.1|69.6|76.9|60.1|81.8|84.1|62.6|68.5|
> > |Yes|Yes|No|Yes|Yes|Yes|51.3|71.3|48.6|60.7|59.4|71.4|75.9|60.1|79.7|84.0|63.0|69.5|
> > |Yes|Yes|Yes|No|Yes|Yes|49.1|69.4|50.9|60.6|59.5|71.8|77.1|61.4|77.8|84.1|62.9|69.5|
> > |Yes|Yes|Yes|Yes|No|No|48.0|69.1|48.3|60.2|56.6|71.0|73.9|60.2|80.6|84.4|61.5|69.0|
> > |Yes|Yes|Yes|Yes|Yes|No|49.17|70.17|50.1|60.1|57.9|70.1|75.6|60.6|81.4|85.9|62.8|69.3|
> > |Yes|Yes|Yes|Yes|No|Yes|48.5|69.1|48.9|59.7|57.1|69.6|76.9|60.1|81.8|84.1|62.6|68.5|
> > |Yes|Yes|Yes|Yes|Yes|Yes|51.03|70.2|52.6|60.9|59.2|70.2|78.6|61.7|83.4|86.1|65.0|69.8|
> >
> > > **weaknesses**: Mismatch of sACC in table 2
> > > **requested changes** Explain the mismatch between Tab. 2 and Tab. 3
> >
> > We would like to clarify that the results in Table 2 of our paper are on the constructed candidate sets for training images. We apologize for this confusion, and have made it clear in the revision. As indicated in Sec3.2 of the revision, our goal was to demonstrate that the K-NN based candidate set achieves higher sACC, indicating it provides a better initialization compared to MLLM-generated labels or a random candidate set. Table 3 shows results on the test set, and hence has differing numbers from Table 2.
> >
> > > Increase in shots, decrease in cACC
> >
> > We understand the concern; however, this drop in cACC with increasing number of shots is observed across all baselines due to the larger MLLM-generated label space. As shown in Fig.2, NeaR consistently outperforms others at all m. For example, at m=10, NeaR-LLaMA achieves 61.8 cACC, vs 54.7 (CoOp-LLaMA) and 52.2 (FineR). NeaR is more robust with increasing m with only a 3.5% drop from m=3 compared to ~6% for others.
> >
> > > Different CLIP Architectures
> >
> > We present results using ViT-L and ViT-G backbones below. We observe performance gains of 5.5% in cACC and 3.2% in sACC with ViT-L, and gains of 6.9% in cACC and 2.9% in sACC with ViT-G. Our method still outperforms CoOp-LLaMA by 3.2% with ViT-L and 4.6% with ViT-G, demonstrating its effectiveness across stronger backbone architectures. We also experiment with the SigLIP ViT-B/16 backbone which has a different pretraining objective compared to CLIP. Our results indicate that our method does not transfer effectively to SigLIP. We conjecture that our method fails because our final objective resembles the InfoNCE objective used by CLIP, and is different from the Sigmoidal loss employed by SigLIP. However, we are competitive with the CoOp baseline in avg cACC. We have added this discussion in Appendix Sec A8.7.
> >
> > | Method| Bird || Car || Dog || Flower | | Pet ||Avg|Avg|
> > |-|-|-|-|-|-|-|-|-|-|-|-|-|
> > || cACC|sACC|cACC|sACC|cACC|sACC|cACC|sACC|cACC|sACC|cACC|sACC|
> > |CoOp-LLaMA (ViT-L)|55.8|73.4|51.1|63.0|65.1|74.3|81.7|63.3|83.2|87.6|67.3|72.3|
> > |NeaR-LLaMA (ViT-L)|54.5|73.1|62.5|62.3|67.9|74.6|82.4|64.7|84.9|87.3|70.5|72.4|
> > |CoOp-LLaMA (ViT-G)|56.2|74.0|51.2|63.1|64.7|73.8|82.2|63.1|82.4|87.5|67.3|72.3|
> > |NeaR-LLaMA (ViT-G)|59.0|74.9|63.0|61.7|66.2|74.2|87.5|64.9|83.7|85.0|71.9|72.1|
> > |CoOp-LLaMA (SigLIP)|36.8|64.1|61.9|62.8|58.5|70.2|71.1|57.2|75.6|80.0|60.8|66.9|
> > |NeaR-LLaMA (SigLIP)|34.0|63.0|71.5|62.9|57.7|70.2|70.9|56.7|72.9|78.6|61.4|66.3|
> >
> > > OOM Issue
> >
> > In Table11 we designed an experiment where candidate sets are extracted directly from the MLLM. For the Car-196 dataset, the generated candidate labels were highly diverse—resulting in a union of approximately 1200 unique labels across all images which leads to OOM issues during text feature computation. As a workaround for the Car-196 dataset alone, we limited the candidate sets to the top-2 labels per image and add those results to Appendix Sec A8.5 in the revised manuscript.
> >
> > > Claim that we are the first to use MLLMs
> >
> > We respectfully clarify that while prior works such as FineR and CaSED do utilize vision-language models like BLIP, their use is fundamentally different from ours. FineR uses BLIP primarily to generate captions or descriptions for subsequent retrieval-based methods, rather than directly predicting class labels. CaSED includes an ablation where BLIP is used to generate candidate labels, but this is only one part of a broader study, and it is not the core of their method.
> > In contrast, our work is tries to explore the use of MLLMs such as GPT-4V or Gemini systematically for generating fine-grained class label predictions at scale. In the revised version, to more clearly reflect our contribution, we clarify that rather than being the first to 'use MLLMs' in any form, our work differs from existing methods by exploring how contemporary models can be leveraged to build a cost-efficient, vocabulary-free fine-grained visual recognition system.

---

> > > ### Author Response · Authors · 2025-07-10
> > > **(3) Cont. Response to Reviewer 6edX**
> > >
> > > > Add space in Related works section
> > >
> > > Thanks for the suggestion. In the revised manuscript, we have added spaces in the Related Works section for better readability.
> > >
> > > > Cost estimates
> > >
> > > We thank the reviewer for the insightful comment. We acknowledge that our cost estimate (~$100 for 32k images) could be improved by leveraging batched inference APIs, which can indeed reduce the cost by up to 5× and ensure timely completion.
> > >
> > > Our original estimate was based on standard per-image API usage (e.g. GPT-4 or Gemini Pro), which many users adopt by default due to ease of integration. However, we agree that using optimized batch pipelines is a more efficient option, potentially reducing the cost to approximately $20 for 32k images. Even by using optimized batched pipeline, our method always requires lesser costs. We add this discussion to Appendix Sec A8.1 in the revision.
> > >
> > > > Table 2 MLLM
> > >
> > > Thanks for pointing out. Unless otherwise stated, we use LLaMA-3.2. We will make this point clear in the revised manuscript.
> > >
> > > > Validity of k=3
> > >
> > > We thank the reviewer for raising this insightful question. To generalize this idea, we conduct an experiment where m is randomly sampled between 1 and 10 ([Random sampling of Flower-102 dataset](https://anonymous.4open.science/r/NeaR-DC02/flowers.png)), and we ablate over different values of k ranging from 1 to 9. We find that the cACC remains relatively stable across this range, with only minor variations. This indicates that our method is not particularly sensitive to the choice of k, which can be selected flexibly. In our implementation, we use k=3 as a practical default. The results for this ablation on the Flowers-102 dataset are shown below. We have added this in Sec A8.11 in the revised manuscript.
> > > | Method| Flower-102 ||
> > > |-|-|-|
> > > ||cACC|sACC|
> > > |k=1|69.8|50.1|
> > > |k=2|72.3|51.7|
> > > |k=3|73.5|52.0|
> > > |k=4|74.7|53.9|
> > > |k=5|74.0|53.1|
> > > |k=6|74.3|53.6|
> > > |k=7|73.6|53.8|
> > > |k=8|71.5|52.6|
> > > |k=9|70.9|52.0|
> > >
> > > > Qualitative results
> > >
> > > When comparing sACC, we observe that MLLMs (GPT4/Geminipro/LLaMA) produce stronger labels (sACC:72.4/73.3/69.9) or predictions than FineR (sACC: 64.3). Therefore, we focus our comparisons on NeaR against MLLMs. Moreover, since our method builds directly on top of noisy outputs from MLLMs, it is most appropriate and fair to benchmark against them. We have added this to Sec4.4 in the revised manuscript.

---

> > > > ### Comment · Reviewer_6edX · 2025-07-21
> > > >
> > > > I thank the authors for their extensive response. While I am generally satisfied with the rebuttal, I still believe that having the sampling strategy collect samples for all the classes could partially game the performance of the models in terms of cluster accuracy. I understand the authors follow the protocol outlined in FineR, and I do not directly attribute the issue to them.
> > > >
> > > > More importantly, one of my requests was to test how random sampling (without any assumption about the presence of all classes in the selection process) would affect the performance of the method. This experiment is missing from the rebuttal. Instead, the authors proposed another version of the same experiment they performed in the main paper (which I criticised) for the imbalanced training setting, where they replaced the range of potential samples for each class from 3-10 to 1-10. First, I do not see how the new setting would be now considered effectively "long-tailed" compared to before. Second, I believe that adding the requested experiment would have demonstrated that the method does not exploit any knowledge of the number of target names (even if indirectly) during the fine-tuning process.

---

> > > > > ### Author Response · Authors · 2025-07-22
> > > > > **Response to Reviewer 6edX**
> > > > >
> > > > > We thank the reviewer for going through our response and for being generally satisfied with the rebuttal. We investigate two aspects of the reviewer's concerns: i) the possibility of gaming cACC scores by implicitly estimating the number of true classes, ii) the performance of NeaR when some class images are missing from the training set.
> > > > >
> > > > > To address the gaming concern, we refer to two experiments. The first is the study on long-tail distribution (Table 6) in the main paper, where the model cannot guess true number of classes, yet the performance is comparable to the m=3 setting. The second is the study on random data distribution, which does not assume a long-tailed class structure (as seen in this image of class distributions [here](https://anonymous.4open.science/r/NeaR-DC02/random_sampling.png) ). This setting is more challenging, as some classes have very few samples. The number of true classes cannot be inferred in our random distribution experiment, and we believe there is no scope for gaming cACC in this setting too. Using m=1 (instead of m=0 as suggested) has no impact on studying the possibility of gaming cACC scores, since the true class names are never revealed and the model can only see the MLLM generated labels. We choose to have at least one training image per class to ensure that the training images are in-distribution. In both these experiments, NeaR outperforms the CoOp baseline by 5.7\% in avg cACC for the long-tail expt (Table 6 in main paper), and by 8.3\% in avg cACC for the random distribution experiment (Table 1 in response).
> > > > >
> > > > > To address the second concern -- we study the performance of NeaR when certain classes are absent from the training set by extending the random distribution experiment to sample m $\in$ [0,10]. The results are shown in the first two rows of the table below. We observe a drop of 2.1% in cACC for NeaR and 1.5% for CoOp-LLaMA compared to the original random experiment (with m$\in$[1,10]). The performance drop is expected as the training images for a few classes are missing, and are now out-of-distribution for the CLIP model. Furthermore the label set during inference cannot include these missing class names. Despite this drop, NeaR still outperforms CoOp-LLaMA by 7.7% in cACC, showing its robustness.
> > > > >
> > > > > To further test the method under more severe missing-class conditions, we also evaluated a worst-case scenario in which we deliberately exclude 25% of the classes from the training set. The results show a further drop in performance, as expected. We emphasize that this issue is not unique to NeaR — even existing methods such as CoOp and FineR would suffer from missing class supervision.
> > > > >
> > > > >
> > > > > | Method| Bird-200 || Car-196 || Dog-120 || Flower-102 | | Pet-37 ||Avg|Avg|
> > > > > |-|-|-|-|-|-|-|-|-|-|-|-|-|
> > > > > || cACC|sACC|cACC|sACC|cACC|sACC|cACC|sACC|cACC|sACC|cACC|sACC|
> > > > > |CoOp-LLaMA (random)|41.4|52.1|42.0|56.5|42.1|53.4|56.6|42.2|50.9|55.0|46.6|51.8|
> > > > > NeaR-LLaMA (random)|42.4|52.2|46.9|55.8|50.9|54.0|68.7|50.7|62.8|52.5|54.3|53.0|
> > > > > |CoOp-LLaMA (25% missing)|39.9|51.1|39.8|56.3|40.6|52.4|54.0|40.5|49.8|52.1|44.8|50.5|
> > > > > NeaR-LLaMA (25% missing)|38.5|49.9|43.3|55.6|47.7|52.1|63.0|48.6|59.3|50.6|50.4|51.3|

---

### Author Response · Authors · 2025-07-08
**Acknowledgement of Reviews**

Dear Reviewers,

We are thankful for the thoughtful and constructive feedback. We are in the process of drafting a rebuttal. Our response is delayed by a few days to ensure we can address each comment properly. We will submit our rebuttal, and a revised version of the manuscript, within the next 48 hours. Thank you for understanding.


Best Regards,
The Authors.

---

### Author Response · Authors · 2025-07-10
**Common Response to Reviewers**

We thank all reviewers for their thoughtful feedback. We are pleased to see the following encouraging comments from the reviewers:

1) The work is well-motivated and tackles a novel problem that is more practical than traditional zero-shot fine-grained visual recognition (6edX, xzFU).
2) The proposed method shows improved efficiency and performance compared to the existing works (Q8Kb, 6edX).
3) The proposed method is straightforward to implement (Q8Kb).
4) The presentation is curated, and the content is clearly explained. (6edX)

As a summary, while recent Multimodal Large Language Models (MLLMs) show potential for VF-FGVR, querying these models for each test input is impractical because of high costs and prohibitive inference times. To address these limitations, we introduce Nearest-Neighbor Label Refinement (NeaR), a novel approach that fine-tunes a downstream CLIP model using labels generated by an MLLM. Our approach constructs a weakly supervised dataset from a small, unlabeled training set, leveraging MLLMs for label generation. NeaR is designed to handle the noise, stochasticity, and open-endedness inherent in labels generated by MLLMs, and establishes a new benchmark for efficient VF-FGVR.

We made the following changes to the main paper of the revised manuscript.
1) We added thorough ablation of each component across all datasets to Sec 4.2.
2) We slightly modify the claim that we are the first to use MLLMs and is reflected in Sec 1.
3) We add a discussion on comparison with a retrieval-augmented method, along with other pseudolabel approaches to Sec 2.
4) We add other clarifications as requested.

We include additional experiments as requested in the Appendix: discussion on clustering accuracy is added to Appendix Sec A8.11, discussion on Random sampling of training images is added to Appendix Sec A8.2, discussion on Candidate sets constructed using different backbones is added to Appendix Sec A8.5, discussion on different CLIP architectures is added to Appendix A8.7, discussion on the OOM issue is added to Appendix A12, discussion on cost estimates is added to Appendix A8.1, discussion on validity of k=3 is added to Appendix A8.11, discussion on with and without prompt tuning methods is added to Sec A8.9, discussion with other pseudo-labeling methods such as DualCoOp and VLPL is added to Sec A8.10, discussion on Visualizing noisy samples from GMM is added to Sec A8.16, discussion on choice of vision encoder for candidate set construction is added to Sec A8.6.

---

### Decision · Action_Editor_M7ea · 2025-07-25

**Recommendation:** Accept with minor revision

**Additional Comments:**

I think the paper is mostly acceptable. Some minor modifications can improve the paper.

1. Some citations are wrongly formatted (e.g, the revised texts). Please use \citep instead of \cite
2. Many important experiments are in the Appendix. Since TMLR has no page limit, I strongly recommend moving them to the main paper to make the paper self-contained.
3. (Minor) Since TMLR has no page limit, please consider removing `vspace`. For example, Section 4.1 and the previous paragraphs seem to have `vspace`. (ignore this comment if there is no vspace)
4. (Minor) I think that it'd be better to handle the concern of Reviewer 6edX. For example, as far as the AC understood, it is not clearly stated that "Our experimental protocol strictly follows that of FineR, using three images per class as the default setting and adopting the same evaluation metrics (cACC and sACC)" (from the rebuttal comment). A novice author might be confused about the setting. I think it'd be better to clarify that the setting used in this paper is built upon the existing paper. (if any paragraph already clarifies this, please ignore this comment. However, at first glance, I couldn't find relevant information)

**Audience:**

Yes

**Audience Explanation:**

This paper explores a new direction: Vocabulary-Free Fine-grained Visual Recognition (VF-FGVR). Although, as one of the reviewers pointed out, the current setting could be somewhat artificial, I think that we can develop more ideas based on this attempt.

**Claims And Evidence:**

Yes

**Claims Explanation:**

This paper introduces a new task, named Vocabulary-Free Fine-grained Visual Recognition (VF-FGVR). The main claims made by the paper are:

- MLLMs can solve VF-FGVR, but they cost a lot
- The proposed method (Nearest-Neighbor Label Refinement (NeaR)) can solve the problem with less cost than MLLMs

Considering that the main focus of this paper is efficiency, I think this paper shows great empirical results.

In terms of the method design choice, as the reviewers pointed out, not all design choices are clear. All reviewers asked for additional ablation studies to justify the design choice (e.g., backbone, `k` for KNN, label refinement design choices, ...). During the rebuttal period, the authors provided requested ablation studies to address the concern. Although there is still some ambiguity regarding the design choice details, I think the current contribution is sufficient.

It is also worth noting that one reviewer pointed out the validity of the proposed experimental setting. Namely, the setting goes partially against the idea of vocabulary-free, because we would need prior information about the number of samples or classes. However, the reviewer also pointed out that the authors did not formulate the task, so they don't consider the issue a rejection case. I partially agree with the argument by the reviewer. The current formulation can be somewhat artificial and cannot be extended to a more general setting. However, I think it is also worth exploring diverse and various directions of machine learning, and the problem formulation is somewhat acceptable to TMLR.

---

> ### Author Response · Authors · 2025-08-21
> **Camera-Ready Submission**
>
> We once again thank all reviewers and the AE for the constructive feedback, which has helped improve the presentation of our paper. As suggested by the AE, we moved all the important experiments from the Appendix to the main paper, corrected the citations, and avoided the usage of \vspace.
>
> For completeness, we also summarize here the changes made in earlier versions and after reorganizing the paper:
> 1) We slightly modify the claim that we are the first to use MLLMs, as reflected in Sec 1.
> 2) We added a discussion on comparison with a retrieval-augmented method, and other pseudolabel approaches to Sec 2.
> 3) We added a thorough ablation of each component across all datasets to Sec 4.2.
> 4) Discussion on Random sampling of training images is added to Sec 4.2.
> 5) Discussion on the validity of k=3 is added to Sec 4.3.
> 6) Discussion on the choice of the vision encoder for candidate set construction is added to Sec 4.4.
> 7) Discussion on different CLIP architectures is added to Sec 4.5.
> 8) Discussion on the Adapter-based Finetuning method is added to Sec 4.6.
>
>
> We include additional experiments as requested in the Appendix-- discussion on cost estimates is added to Appendix A8.1; discussion on clustering accuracy is added to Appendix Sec A8.7; discussion on the OOM issue is added to Appendix A8.4; discussion with other pseudo-labeling methods such as DualCoOp and VLPL is added to Sec A8.9; discussion on Visualizing noisy samples from GMM is added to Sec A8.11.